# Correcting for Model Changes in Statistical Postprocessing – An approach based on Response Theory

Jonathan Demaeyer [1,2] and Stéphane Vannitsem [1,2]

[1]Institut Royal Météorologique de Belgique, Avenue Circulaire, 3, 1180 Brussels, Belgium
[2]European Meteorological Network, Avenue Circulaire, 3, 1180 Brussels, Belgium

*Correspondence to:* jodemaey@meteo.be

**Abstract.** For most statistical postprocessing schemes used to correct weather forecasts, changes to the forecast model induce a considerable reforecasting effort. We present a new approach based on response theory to cope with slight model changes. In this framework, the model change is seen as a perturbation of the original forecast model. The response theory allows us then to evaluate the variation induced on the parameters involved in the statistical postprocessing, provided that the magnitude of this perturbation is not too large. This approach is studied in the context of simple Ornstein-Uhlenbeck models, and then on a more realistic, yet simple, quasi-geostrophic model. The analytical results for the former case help to pose the problem, while the application to the latter provide a proof-of-concept and assesses the potential performances of response theory in a chaotic system. In both cases, the parameters of the statistical postprocessing used – an *Error-in-Variables* Model Output Statistics (EVMOS) – are appropriately corrected when facing a model change. The potential application in an operational environment is also discussed.

# 1 Introduction

A generic property of the atmospheric dynamics is its sensivity to initial conditions. This implies that probabilistic forecasts will always be needed to adequately describe this behaviour (Wilks, 2011). Indeed, these methods represent a way to go beyond the natural predictability barrier that the chaotic atmospheric models exhibit (Vannitsem, 2017). These forecasts are at the same time subject to the impact of the presence of structural uncertainties, also known as *model errors*. Such errors degrade the forecasts as well, and their impact needs to be mitigated.

*Statistical postprocessing methods* are used to correct the operational predictions of the atmospheric models. An important family of statistical techniques used to postprocess the forecasts are linear regression techniques, with possibly multiple predictors (Glahn and Lowry, 1972; Vannitsem and Nicolis, 2008), also known as Model Output Statistics (MOS). This rather simple but very efficient technique can be adapted to ensemble forecasts (e.g. Vannitsem (2009); Johnson and Bowler (2009); Glahn et al. (2009); Van Schaeybroeck and Vannitsem (2015)). One of the first approaches that was proposed is called *Error-in-Variable* MOS (EVMOS) because it takes into account the presence of errors in both the observations and model observables (Vannitsem, 2009).

Despite their simplicity, most postprocessing schemes depend on the availability of a database of past forecasts, that allows one to "train" the regression algorithm by comparison with the observations database. Operational models are however subject to frequent evolution cycles, which are needed to improve their representation of the atmospheric processes. Therefore, there is a continuous need to recompute forecasts starting from past initial conditions with the latest model version, to avoid a degradation of the postprocessing schemes due to model change. Such recomputation of the past forecasts are called *reforecasts*, and typically requires a huge data storage and management framework, as well as large computational resources (Hamill, 2018). For instance, the *European Center for Medium-range Weather Forecast* (ECMWF) and the *National Weather Service* (NWS) both produce hundreds of reforecasts every week (Hamill et al., 2013).

A recent research has investigated non-homogeneous regression with time-adaptive training scheme, for which a trade-off between large training data sets for stable estimates and the benefit of a shorter training period for faster adaptation to data changes is considered (Lang et al., 2020). These results can help mitigate the impact of model change on postprocessing and may call into question the need for reforecast systems. These systems do however help to better represent rare events, they increase the size of the training data sets and greatly improve sub-seasonal forecasts (Scheuerer and Hamill, 2015; Hamill, 2018), which can justify their prohibitive cost.

The present work investigates another research direction and considers a new technique to reduce the cost of adapting a linear postprocessing scheme to a model change. This method relies on the response theory for dynamical systems (Ruelle, 2009) and assumes that the model variation can be written as analytical perturbations of the model tendencies. In this context, parameters variations as well as new terms in the tendencies are potential model changes.

In section 2, we start by introducing the Ruelle response theory that is used to adapt past postprocessing parameters to new models. A didactical example of such adaptation is considered with simple Ornstein-Uhlenbeck models in section 3. It is used to describe the methodology and the concept involved. We show that obtaining a new postprocessing scheme after a model

change requires the computation of the response of the average of the involved predictors, seen as observables of the system. In the simple case considered, exact analytical results for the response can be obtained at any order. The correction of the model observables and the postprocessing parameters due to the model change only requires the response-theory corrections up to the second order.

In section 4, a more complex case is considered with a toy model of atmospheric variability in the form of a 2-layer quasi-geostrophic model with an orography. We compute the linear response of the predictors of the postprocessing for two model change experiments involving a modification of the friction and the horizontal temperature gradient of the model. The response theory approach provides an efficient correction of the postprocessing scheme up to a lead time of 4 days, which matches the lead-time window where the scheme's correction is efficient.

In the last section, we discuss the implications that this new method could have on operational forecast postprocessing systems, as well as new research avenues to be explored.

## 2   Response theory

The systems used to produce the weather forecasts are typically non-linear dynamical systems whose time evolution is governed by multi-dimensional ordinary differential equations:

$$\dot{\boldsymbol{y}} = \boldsymbol{F}(t, \boldsymbol{y}). \tag{1}$$

The generic chaotic nature of these systems for some parameter values implies that they are sensitive to the initial data used to produce the forecasts. For such chaotic dynamical systems, one can assume that a well-defined time-invariant measure exists, and with which the averages are performed. However, the existence of such measures has been proved for systems that are *uniformly hyperbolic* and they are called *SRB measure* (Young, 2002), but rigorous proofs for other systems are rather difficult

to obtain. A way to proceed is then to proceed as if physical systems were uniformly hyperbolic. This assumption is called the Gallavotti-Cohen hypothesis (Gallavotti and Cohen, 1995a, b). With this assumption, response theory has been succesfully used in various weather and climate-related problems (Demaeyer and Vannitsem, 2018; Vissio and Lucarini, 2018; Lembo et al., 2019; Bódai et al., 2020). Indeed, the systems used to produce weather forecasts are typically not *uniformly hyperbolic*, but thanks to the aforementioned hypothesis, one can still use what will follow and compare with the results obtained with

experiments.[1] It is the rationale behind the formal presentation of the linear response theory for general systems like (1) in Ruelle (1998a). Main concepts that will be used in this article are now introduced.

### 2.1   Perturbations of dynamical systems

We shall assume for simplicity that the system (1) is autonomous and given by

$$\dot{\boldsymbol{y}} = \boldsymbol{F}(\boldsymbol{y}). \tag{2}$$

---

[1]We point the reader to recent articles dealing with the validity of the response theory for weakly hyperbolic systems and time series (Gottwald et al., 2016; Wormell and Gottwald, 2018).

In the general setting considered, let's assume that any given probability measure converges to a unique invariant measure $\rho$ under the time evolution given by the Liouville equation of (2). This measure is used to compute the average of an arbitrary observable $A$ (a smooth function of the state $\boldsymbol{y}$) of the system, which is given by

$$\langle A \rangle_{\boldsymbol{y}} = \int \rho(\mathrm{d}\boldsymbol{y}) \, A(\boldsymbol{y}) \tag{3}$$

and assuming the ergodicity of the system, a time average of the observable $A$ along a trajectory of the system on its attractor can be equivalently performed:

$$\langle A \rangle_{\boldsymbol{y}} = \lim_{T \to \infty} \frac{1}{T} \int_0^T \mathrm{d}\tau \, A(\boldsymbol{y}(\tau)) \tag{4}$$

where $\boldsymbol{y}(\tau)$ is a solution of Eq. (2). If a perturbation $\boldsymbol{\Psi}$ of the dynamical system is introduced in the original system at the time $\tau = 0$:

$$\dot{\boldsymbol{y}} = \boldsymbol{F}(\boldsymbol{y}) + \boldsymbol{\Psi}(\boldsymbol{y}), \tag{5}$$

it induces a perturbation of the observable's average which at first order is given by[2]

$$\delta \langle A(\tau) \rangle_{\boldsymbol{y}} = \int \rho(\mathrm{d}\boldsymbol{y}_0) \, \delta A(\boldsymbol{f}^\tau(\boldsymbol{y}_0)) = \int \rho(\mathrm{d}\boldsymbol{y}_0) \, \boldsymbol{\delta y}(\tau)^\mathsf{T} \cdot \boldsymbol{\nabla}_{\boldsymbol{f}^\tau(\boldsymbol{y}_0)} A \tag{6}$$

where $\boldsymbol{f}^\tau$ is the flow the system (2) mapping an initial condition $\boldsymbol{y}_0$ to the system's state at time $\tau$: $\boldsymbol{y}(\tau) = \boldsymbol{f}^\tau(\boldsymbol{y}_0)$. $\boldsymbol{\delta y}$ is the perturbation of the trajectory of the system induced by the perturbation $\boldsymbol{\Psi}$. This formula gives the transient response to the perturbation, and long time average of the integrand of (6) gives the stationary response to the perturbation, i.e. the sensibility
$\delta \langle A \rangle_{\boldsymbol{y}}$ of the system observables to the perturbation (Eyink et al., 2004; Wang, 2013). The higher-order corrections $\delta^k \langle A(\tau) \rangle_{\boldsymbol{y}}$ can in principle be computed as well but are quite complicate to obtain for chaotic dynamical systems, see for instance Lucarini (2009). We will show an analytically tractable case in Section 3.

## 2.2   The tangent linear model

The linear perturbation $\boldsymbol{\delta y}$ of the trajectories of (2) can be computed by introducing $\boldsymbol{y} + \boldsymbol{\delta y}$ in Eq. (5) to get at first order

$$\dot{\boldsymbol{\delta y}} = \boldsymbol{\nabla}_{\boldsymbol{y}} \boldsymbol{F} \cdot \boldsymbol{\delta y} + \boldsymbol{\Psi}(\boldsymbol{y}) \tag{7}$$

where $\boldsymbol{y}$ is the solution of system (2) and $\boldsymbol{\nabla}_{\boldsymbol{y}} \boldsymbol{F}$ is the Jacobian matrix evaluated along this solution. Therefore, both Eqs. (2) and (7) have to be integrated simultaneously. In the weather forecasting context, this latter linearised equation is called the tangent linear model of system (2) (Kalnay, 2003) *plus* the perturbation term $\boldsymbol{\Psi}$. Without this latter term, it provides the
25 linearised time evolution of a perturbation $\boldsymbol{\delta y}$ superimposed to the initial conditions. Here, Eq. (7) is initialised with $\boldsymbol{\delta y}(0) = 0$ and provides the linear "response" of the trajectory $\boldsymbol{y}(\tau)$ to the perturbation $\boldsymbol{\Psi}$. It is thus assumed that there is no interference

---

[2]When taking the gradient of a function $A$, the notation $\boldsymbol{\nabla}_{\boldsymbol{y}} A$ means taking the gradient at the point $\boldsymbol{y}$, i.e. evaluating $\boldsymbol{\nabla}_{\boldsymbol{y}} A(\boldsymbol{y})$.

due to initial condition errors in the perturbation problem. Note however that the effects on the trajectories of both the initial conditions perturbation and the $\boldsymbol{\Psi}$ perturbation can be investigated through this equation by setting $\boldsymbol{\delta y}(0) \neq 0$, altough we are not aware of any study of the response to both type of perturbations together.

The tangent model provides thus the tool through which we will evaluate the impact of the model change on the average used by statistical postprocessing schemes. In other words, the tangent model will allow us to take into account the information on the model change (viewed as a perturbation of the initial model) to modify the previous postprocessing scheme and adapt it to the new model. The solution to Eq. (7) with $\boldsymbol{\delta y}(0) = 0$ is given by

$$
\boldsymbol{\delta y}(\tau) = \int_0^\tau \mathrm{d}\tau' \, \mathbf{M}\left(\tau - \tau', \boldsymbol{f}^{\tau'}(\boldsymbol{y_0})\right) \cdot \boldsymbol{\Psi}\left(\boldsymbol{f}^{\tau'}(\boldsymbol{y_0})\right),
\tag{8}
$$

where $\mathbf{M}$ is the fundamental matrix of solutions of Eq. (7) (Gaspard, 2005; Nicolis, 2016):

$$
\mathbf{M}(\tau, \boldsymbol{y}) = \boldsymbol{\nabla_y} \boldsymbol{f}^\tau
\tag{9}
$$

solution of the homogeneous equation $\dot{\mathbf{M}} = \boldsymbol{\nabla_y} \boldsymbol{F} \cdot \mathbf{M}$. Using the chain rule, the response (6) is rewritten in term of the perturbation alone:

$$
\delta\langle A(\tau)\rangle_{\boldsymbol{y}} = \int_0^\tau \mathrm{d}\tau' \int \rho(\mathrm{d}\boldsymbol{y_0}) \, \boldsymbol{\Psi}\left(\boldsymbol{f}^{\tau'}(\boldsymbol{y_0})\right)^\top \cdot \boldsymbol{\nabla}_{\boldsymbol{f}^{\tau'}(\boldsymbol{y_0})} A\left(\boldsymbol{f}^\tau(\boldsymbol{y_0})\right)
\tag{10}
$$

where the causality of the perturbation acting on the system and perturbing the averaged observable appears (Lucarini, 2008), since $\tau' < \tau$. We will also use this alternative expression throughout the article. Note that when the initial perturbation $\boldsymbol{\delta y}(0)$ is not equal to 0, additional terms to Eqs. (8) and (10) will appear. These will not be addressed here but some hints can be found in Nicolis et al. (2009) and Nicolis (2016).

## 2.3 Non-stationary response theory

The equation (6) gives the transient, non-stationary response to the perturbation, evaluated for averages computed with the invariant measure. However, in this work, we need to evaluate response to perturbation for averages computed with non-stationary measures evolving in time. In that sense, it is a *non-stationary response theory*, done for arbitrary initial probability density. As such, all the formula presented are valid if the measure being used is the measure at the time when the perturbation is introduced ($\tau = 0$), as shown in Appendix A. In this case, other usual formula obtained through substitution, for instance to obtain an adjoint representation of (10), should be used with care since the measure is no longer invariant and an extra Jacobian term appears in the integrand.

We will thus also assume that the measures $\rho_\tau$ being used are absolutely continuous with respect to the Lebesgue measure. In this case, we can write $\rho_\tau(\mathrm{d}\boldsymbol{y}) = \rho_\tau(\boldsymbol{y}) \, \mathrm{d}\boldsymbol{y}$. We now present the problem of model change in the framework of postprocessing and show on a simple stochastic model[3] how response theory allows to tackle the issue.

---

[3]Response theory is also valid for stochastic models with a well-defined stationary measure, as shown by Lucarini (2009).

## 3 A simple analytical example

In order to get a first impression of the impact of a model change on a postprocessing scheme, we consider two Ornstein-Uhlenbeck processes representing the reality $x(\tau)$ and a model $y(\tau)$ of the reality. These processes obey the following equations:

$$\dot{x}(\tau) = -\lambda_x\, x(\tau) + K_x + Q_x\, \xi_x(\tau) \tag{11}$$

$$\dot{y}(\tau) = -\lambda_y\, y(\tau) + K_y + Q_y\, \xi_y(\tau) \tag{12}$$

where $\xi_x$ and $\xi_y$ are Gaussian white noise processes such that

$$\langle \xi_x(\tau) \rangle = \langle \xi_y(\tau) \rangle = 0$$

$$\langle \xi_x(\tau)\, \xi_x(\tau') \rangle = \delta(\tau - \tau')$$

$$\langle \xi_y(\tau)\, \xi_y(\tau') \rangle = \delta(\tau - \tau')$$

$$\langle \xi_x(\tau)\, \xi_y(\tau') \rangle = 0$$

These are therefore uncorrelated Ornstein-Uhlenbeck processes with noise amplitudes $Q_x$ and $Q_y$.

We then consider a change $\Psi_y$ of the model $y(\tau)$, possibly improving or degrading the forecast performances:

$$\dot{\hat{y}}(\tau) = -\lambda_y\, \hat{y}(\tau) + K_y + Q_y\, \xi_y(\tau) + \Psi_y(\tau) \tag{13}$$

where

$$\Psi_y(\tau) = -\kappa\left(\delta K + \delta Q\, \xi_y(\tau)\right) \tag{14}$$

with $\delta K = K_y - K_x$ and $\delta Q = Q_y - Q_x$. It can represent, for example, a better parameterisation of subgrid-scale processes or an increase of the model resolution. Note that the best correction is obtained if $\kappa = 1$.

We have thus the reality $x(\tau)$ and two different models of it: $y(\tau)$ and $\hat{y}(\tau)$. We now want to evaluate the difference between a postprocessing scheme constructed before the model change (with the past forecasts of the model $y(\tau)$), and one constructed after (with the past forecasts of model $\hat{y}(\tau)$).

### 3.1 The postprocessing method

We now consider a forecast situation where the model $y$ is initialised at the time $\tau = 0$ with a perfect observation of the reality: $y(0) = x(0) = x_0$. We use the *Error-in-Variables Model Output Statistics* (EVMOS) postprocessing scheme (Vannitsem, 2009) to correct the forecasts of the model $y$ based on these initial conditions. In this context, given $N$ past forecasts $y_n$ and

observations $x_n$, the correction of the univariate EVMOS postprocessing of variable $x$ from a new forecast $y(\tau)$ is provided by the linear regression

$$y_C(\tau) = \alpha(\tau) + \beta(\tau)\,y(\tau) \tag{15}$$

The coefficients $\alpha$ and $\beta$ are obtained by minimising the functional

$$J(\tau) = \sum_{n=1}^{N} \frac{[\{\alpha(\tau) + \beta(\tau)y_n(\tau)\} - x_n(\tau)]^2}{\sigma_x^2(\tau) + \beta^2(\tau)\,\sigma_y^2(\tau)}, \tag{16}$$

and are thus given by the equations:

$$\alpha(\tau) = \langle x(\tau)\rangle - \beta(\tau)\,\langle y(\tau)\rangle \tag{17}$$

$$\beta(\tau) = \sqrt{\frac{\sigma_x^2(\tau)}{\sigma_y^2(\tau)}} \tag{18}$$

where

$$\sigma_x^2(\tau) = \left\langle \left(x(\tau) - \langle x(\tau)\rangle\right)^2 \right\rangle \tag{19}$$

$$\sigma_y^2(\tau) = \left\langle \left(y(\tau) - \langle y(\tau)\rangle\right)^2 \right\rangle \tag{20}$$

The averages $\langle \cdot \rangle$ are taken over an ensemble of past forecasts and observations. This approach has been developed to obtain a correct climatological forecast calibration. It constitutes a simple setting in which the impact of model changes can be evaluated and corrected. More sophisticated approaches can be evaluated in the future (other MOS schemes, ensemble MOS,...).

Since we are dealing with simple analytical models here, we can compute the theoretical values of the coefficient $\alpha$ and $\beta$ with an infinite ensemble of past forecasts, and the averaged quantities involved in this computation are then given by the averages of an infinite number of realisations of the Ornstein-Uhlenbeck processes, as if we had an infinite ensemble of past forecasts.

## 3.2 Averaging the Ornstein-Uhlenbeck processes

For the reality $x$ and the model $y$, we directly get the averages (Gardiner, 2009)

$$\langle x(\tau)\rangle = \langle x_0\rangle e^{-\lambda_x \tau} + \frac{K_x}{\lambda_x}\left(1 - e^{-\lambda_x \tau}\right) \tag{21}$$

$$\sigma_x^2(\tau) = \sigma_{x_0}^2 e^{-2\lambda_x \tau} + \frac{Q_x^2}{2\lambda_x}\left(1 - e^{-2\lambda_x \tau}\right) \tag{22}$$

and

$$\langle y(\tau)\rangle = \langle x_0\rangle e^{-\lambda_y \tau} + \frac{K_y}{\lambda_y}\left(1 - e^{-\lambda_y \tau}\right) \tag{23}$$

$$\sigma_y^2(\tau) = \sigma_{x_0}^2 e^{-2\lambda_y \tau} + \frac{Q_y^2}{2\lambda_y}\left(1 - e^{-2\lambda_y \tau}\right) \tag{24}$$

where we note that the model is initialised with the same initial conditions as the reality:

$$\langle y(0) \rangle = \langle x(0) \rangle = \langle x_0 \rangle \qquad , \qquad \sigma_y^2(0) = \sigma_x^2(0) = \sigma_{x_0}^2 \tag{25}$$

We get the postprocessing coefficients before the model change $\alpha(\tau)$ and $\beta(\tau)$ by inserting these expressions in the equations (17) and (18).

Similarly, we get the same kind of results for the model $\hat{y}$, after the model change $\Psi_y$:

$$\langle \hat{y}(\tau) \rangle = \langle x_0 \rangle e^{-\lambda_y \tau} + \frac{K_y - \kappa \delta K}{\lambda_y} \left( 1 - e^{-\lambda_y \tau} \right) \tag{26}$$

$$\sigma_{\hat{y}}^2(\tau) = \sigma_{x_0}^2 e^{-2\lambda_y \tau} + \frac{(Q_y - \kappa \delta Q)^2}{2\lambda_y} \left( 1 - e^{-2\lambda_y \tau} \right) \tag{27}$$

and we also obtain the postprocessing coefficients after the model change $\hat{\alpha}(\tau)$ and $\hat{\beta}(\tau)$ (see also the analysis in Vannitsem (2011)). We can also compute the variation of the bias $\alpha$:

$$\hat{\alpha}(\tau) - \alpha(\tau) = \delta\alpha(\tau) = \beta(\tau) \langle y(\tau) \rangle - \hat{\beta}(\tau) \langle \hat{y}(\tau) \rangle \tag{28}$$

The ratio between the parameters $\beta$ is given by

$$\frac{\hat{\beta}(\tau)}{\beta(\tau)} = \sqrt{\frac{\sigma_y^2(\tau)}{\sigma_{\hat{y}}^2(\tau)}} \tag{29}$$

For $\tau \gg \max(1/\lambda_x, 1/\lambda_y)$, we note that this ratio tends to

$$\frac{\hat{\beta}(\tau)}{\beta(\tau)} \approx \frac{1}{1 - \kappa \delta Q / Q_y} \tag{30}$$

and the difference between the biases $\alpha$ of the two models is approximatively given by:

$$\delta\alpha(\tau) \approx -\beta(\tau) \frac{K_y}{\lambda_y} \left[ \frac{1 - \kappa \delta K / K_y}{1 - \kappa \delta Q / Q_y} - 1 \right] . \tag{31}$$

Let us now assume that the model change $\Psi_y$ can be considered as a perturbation of the initial model $y$. Using response theory, the averages $\langle \hat{y} \rangle$ and $\sigma_{\hat{y}}^2$ can be estimated using the initial model $y$ instead of the perturbed model $\hat{y}$. In turn, these new estimated averages give us the new postprocessing scheme coefficients $\hat{\alpha}$ and $\hat{\beta}$. We now detail the results obtained by using

this method.

### 3.3 Model Change and Response theory

After the model change, the forecasts are provided by the model $\hat{y}$ and their time-evolution is given by Eq. (13). This model can be seen as a perturbation of the model $y$ by the term $\Psi_y$ given by Eq. (14). In such case, given an observable $A$, its average after the model change can then be related to its original average by

$$\langle A(\tau) \rangle_{\hat{y}} = \langle A(\tau) \rangle_y + \delta\langle A(\tau) \rangle_y + \delta^2 \langle A(\tau) \rangle_y + \dots \tag{32}$$

where the averages on the right-hand side are taken over the forecasts of model $y$. Response theory allows us to obtain the average over the model $\hat{y}$ forecasts (the left-hand side) based solely on the average over the model $y$ forecasts. The $\hat{y}$ model forecasts are therefore not required to estimate the new postprocessing scheme.

The observables depend on the lead time $\tau$ of the forecast, as do the parameters $\alpha$ and $\beta$ which determine the postprocessing correction for every lead time. This reflects the fact that the postprocessing problem is typically a non-stationary initial value problem, since the initial conditions of the model Eqs. (12) and (13) are typically not chosen on their respective model attractor, but rather as observations[4] of the reality (11). As a consequence, the model averages (32) relax toward the stationary response in the long-time limit, and the stationary response theory (Ruelle, 2009; Wang, 2013) cannot provide us their short-time relaxation behaviour. Instead, the Ruelle time-dependent response theory should be used (Ruelle, 1998a). It follows that, if the perturbation (14) is small, then the first order is given by (see section 2):

$$\delta\langle A(\tau)\rangle_y = \int\limits_0^\tau \mathrm{d}\tau' \int \mathrm{d}x_0\,\rho_0(x_0)\left\langle \Psi_y(\tau')\nabla_{f^{\tau'}(x_0)}A\left(f^\tau(x_0)\right)\right\rangle \tag{33}$$

where $\rho_0$ is the distribution of the initial conditions (observations) used to initialise the models. $\nabla_x$ is the gradient evaluated at the point $x$, and here it is the simple derivative. As indicated by Eq. (25), in the postprocessing framework, $\rho_0$ is taken as the stationary/invariant distribution of the reality. As shown in Appendix A, Eq. (33) can be obtained through a Kubo-type perturbative expansion (Lucarini, 2008). We remark that this example deals with stochastic models, due to which we have to perform an additional averaging over the realisations of the stochastic processes, denoted here as $\langle\cdot\rangle$ (Lucarini, 2012). Finally the mapping $f^\tau$ which appears in Eq. (33) is the stochastic flow:

$$f^\tau(x_0) = x_0\, e^{-\lambda_y\tau} + \int\limits_0^\tau \mathrm{d}\tau'\, e^{-\lambda_y(\tau-\tau')}\left[Q_y\,\xi_y(\tau') + K_y\right]. \tag{34}$$

This maps an initial condition $x_0$ of the model $y$ to the state $f^\tau(x_0)$ of a realisation of this model at the later lead time $\tau$. The principle of causality is thus implicit in Eq. (33), which estimates the impact of the perturbation $\Psi_y$ on the subsequent perturbed model time-evolution by developing around the unperturbed model $y$ trajectories.

Evaluating Eq. (33) and its stochastic integrals (Gardiner, 2009) gives us the variation of the averages $\langle y(\tau)\rangle$ and $\langle y(\tau)^2\rangle$ to the perturbation $\Psi_y$:

$$\delta\langle y(\tau)\rangle_y = -\kappa \int\limits_0^\tau \mathrm{d}\tau'\,\delta K\, e^{-\lambda_y(\tau-\tau')} = -\frac{\kappa}{\lambda_y}\,\delta K\left(1 - e^{-\lambda_y\tau}\right) \tag{35}$$

$$\delta\langle y(\tau)^2\rangle_y = -2\kappa\,\delta K \int\limits_0^\tau \mathrm{d}\tau'\left[\langle x_0\rangle\, e^{-\lambda_y(2\tau-\tau')} + \frac{K_y}{\lambda_y}\, e^{-\lambda_y(\tau-\tau')}\left(1 - e^{-\lambda_y\tau}\right)\right] - 2\kappa\,\delta Q\, Q_y \int\limits_0^\tau \mathrm{d}\tau'\, e^{-2\lambda_y(\tau-\tau')}$$
$$\hspace{3cm} = -2\kappa\,\frac{\delta K}{\lambda_y}\,\langle y(\tau)\rangle\left(1 - e^{-\lambda_y\tau}\right) - \frac{\kappa}{\lambda_y}\,\delta Q\, Q_y\left(1 - e^{-2\lambda_y\tau}\right) \tag{36}$$

---

[4]Here we consider that the observation are perfectly assimilated in the models, and that there is no observation errors. However in operational setups, such errors are of course to be taken into account.

Rearranging these two terms, we also get the following expression for the variation of the variance (24):

$$\delta\sigma_y^2(\tau) = -\frac{\kappa}{\lambda_y}\delta Q\, Q_y\left(1 - e^{-2\lambda_y\tau}\right) - \frac{\kappa^2}{\lambda_y^2}\delta K^2\left(1 - e^{-\lambda_y\tau}\right)^2 \tag{37}$$

Note that the variation (35) corresponds to the exact difference between the average of the two models $\langle\hat{y}(\tau)\rangle - \langle y(\tau)\rangle$. On the other hand the variation given by Eq. (37) lacks the term of order $\kappa^2$ involving $\delta Q$ that appears in the exact difference $\sigma_{\hat{y}}^2(\tau) - \sigma_y^2(\tau)$ given by Eqs. (24) and (27). Instead, another term of order $\kappa^2$ and involving $\delta K$ is present, indicating that higher-order terms of response theory need to be considered to correct it (Ruelle, 1998b). The second-order term is given by the expression[5] (Lucarini, 2012):

$$\delta^2\langle A(\tau)\rangle_y = \int_0^\tau \mathrm{d}\tau' \int_{\tau'}^\tau \mathrm{d}\tau'' \int \mathrm{d}y\,\rho_0(x_0)\left\langle\Psi_y(\tau')\nabla_{f^{\tau'}(x_0)}\Psi_y(\tau'')\nabla_{f^{\tau''}(x_0)}A\left(f^\tau(x_0)\right)\right\rangle. \tag{38}$$

Applying this to the first moment of the $y$ models directly yields

$$\delta^2\langle y(\tau)\rangle_y = 0. \tag{39}$$

On the other hand, integrating the stochastic integrals present in this expression for the moment $\langle y(\tau)^2\rangle$ gives

$$\delta^2\langle y(\tau)^2\rangle_y = \frac{\kappa^2\,\delta K^2}{\lambda_y^2}\left(1 - e^{-\lambda_y\tau}\right)^2 + \frac{\kappa^2\,\delta Q^2}{2\lambda_y}\left(1 - e^{-2\lambda_y\tau}\right) \tag{40}$$

which corrects the $\kappa^2\delta K^2$ term in Eq. (37) and makes the response theory up to order 2 exactly match the difference $\sigma_{\hat{y}}^2(\tau) - \sigma_y^2(\tau)$, for every lead time $\tau$. In fact, the subsequent orders of the response vanish due to the linearity of the simple Ornstein-Uhlenbeck models, which enables us to truncate the response Kubo-like expansion to the second order. Finally, this shows that the (non-stationary) response theory can be used to estimate the postprocessing parameters after the model change based on the forecasts of the initial model. Indeed, instead of the averages $\langle\hat{y}(\tau)\rangle$ and $\langle\sigma_{\hat{y}}^2(\tau)\rangle$, the approximate averages $\langle y(\tau)\rangle + \delta\langle y(\tau)\rangle_y$ and $\sigma_y^2(\tau) + \delta\sigma_y^2(\tau) + \delta^2\sigma_y^2(\tau)$ can be used to compute $\hat{\alpha}$ and $\hat{\beta}$. We emphasise that the second order contribution had to be considered in order to obtain the exact result. Nevertheless, the difference between the first and the second order response is of order $\kappa^2$, which implies that for a small perturbation (model change), the first order will generally be a sufficiently good approximation. A more detailed derivation of the results obtained in this section can be found in the supplementary material.

In order to investigate this research avenue on a case closer to those encountered in reality, we will now consider the application of postprocessing and response theory to a low-order atmospheric model displaying chaos.

## 4 Application to a low-order atmospheric model

A 2-layer quasi-geostrophic atmospheric system on a $\beta$-plane with an orography is considered (Charney and Straus, 1980; Reinhold and Pierrehumbert, 1982). This spectral model possesses well-identified large-scale flow regimes, such as *zonal*

---

[5]This expression is equivalent to the second term of Eq. (1) in Lucarini (2012) upon a time transformation. It can also be obtained by computing explicitly the second order perturbation of the average in Eq. (A14) in Appendix A.

and *blocked* regimes. The horizontal adimensionalised coordinates are denoted $x$ and $y$, the model's domain being defined by $(0 \leq x \leq \frac{2\pi}{n}, 0 \leq y \leq \pi)$, with $n = 2L_y/L_x$ the aspect ratio between its meridional and zonal extents $L_y$ and $L_x$. The two main fields of this model are the 500 hPa pressure anomaly and temperature, which are proportional to the barotropic streamfunction $\psi(x,y)$ and the baroclinic streamfunction $\theta(x,y)$, respectively. Both fields are defined in a zonally periodic channel with no-flux boundary conditions in the meridional direction ($\partial \cdot /\partial x \equiv 0$ at $y = 0, \pi$). The fields are expanded in Fourier modes respecting these boundary conditions:

$$
\begin{aligned}
F_1(x,y) &= \sqrt{2}\cos(y), \\
F_2(x,y) &= 2\cos(nx)\sin(y), \\
F_3(x,y) &= 2\sin(nx)\sin(y), \\
F_4(x,y) &= \sqrt{2}\cos(2y), \\
&\vdots
\end{aligned}
$$

such that

$$
\nabla^2 F_i(x,y) = -a_i^2 F_i(x,y) \tag{41}
$$

with eigenvalues $a_1^2 = 1$, $a_2^2 = a_3^2 = 1 + n^2$, $a_4^2 = 4, \dots$ . We have thus the following decomposition

$$
\psi(x,y) = \sum_{i=1}^{n_a} \psi_i \, F_i(x,y) \tag{42}
$$

$$
\theta(x,y) = \sum_{i=1}^{n_a} \theta_i \, F_i(x,y). \tag{43}
$$

where $n_a$ is the number of modes of the spectral expansion. The partial differential equations controlling the time evolution of the fields $\psi(x,y)$ and $\theta(x,y)$ can then be projected on the Fourier modes to finally give a set of ordinary differential equations for the coefficients $\psi_i$ and $\theta_i$:

$$
\dot{\boldsymbol{x}} = \boldsymbol{F}(\boldsymbol{x}) \quad , \quad \boldsymbol{x} = (\psi_1, \dots, \psi_{n_a}, \theta_1, \dots, \theta_{n_a}) \tag{44}
$$

that can be solved with usual numerical integrators. All variables are adimensionalised. The ordinary differential equations of the model are detailed in Appendix B.

In the version proposed by Reinhold and Pierrehumbert using the 10 first modes, beyond a certain value of the zonal temperature gradient, the system displays chaos and makes transitions between the blocked and zonal flow regimes embedded in its global attractor. Here, we use their main adimensionalised parameters values: the friction at the interface between the two layers $k_d = 0.1$, the friction at the bottom surface $k'_d = 0.01$, and the aspect ratio of the domain $n = 1.3$. The $\beta$-plane lies at mid-latitude (50°) and the Coriolis parameter $f_0$ is set accordingly.

In the present work, the parameter $h_d$, the Newtonian cooling coefficient is fixed to $0.3$ instead of the value found in Reinhold and Pierrehumbert (which is $h_d = 0.045$). Two additional fields have to be specified on the domain: $\theta^\star(x,y)$, the radiative

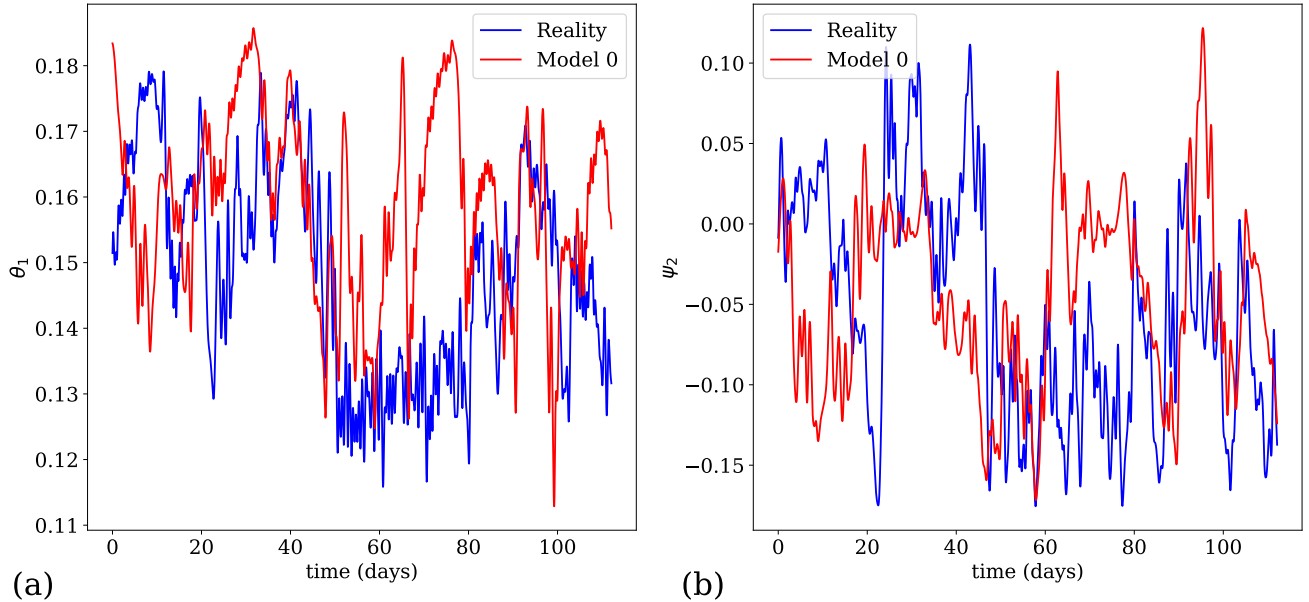

**Figure 1.** Dynamics of reference system and model 0 of the postprocessing experiment with modification of the friction coefficient (see Table 1), for: (a) time evolution of the variable $\theta_1$, (b) time evolution of the variable $\psi_2$.

equilibrium temperature field, and $h(x,y)$, the topographic height field. These fields can be decomposed by projecting them onto the eigenfunctions of the Laplacian as before. The corresponding coefficients $\theta_i^\star$ and $h_i$ then allow for writing these fields as sums of weighted eigenfunctions:

$$\theta^\star(x,y) = \sum_{i=1}^{n_a} \theta_i^\star F_i(x,y) \tag{45}$$

$$5 \quad h(x,y) = \sum_{i=1}^{n_a} h_i F_i(x,y). \tag{46}$$

In the present case, we consider that the only non-zero coefficients are $\theta_1^\star = 0.2$ and $h_2 = 0.4$, meaning that the radiative equilibrium profile is given by the zonally varying function $\sqrt{2}\cos(y)$ and the orography is made of a mountain and a valley shaped by the function $2\cos(nx)\sin(y)$. Again, the value of the temperature gradient $\theta_1^\star$ is larger than the one chosen in Reinhold and Pierrehumbert (which is $\theta_1^\star = 0.1$) to increase the chaotic variability in the system. Trajectories of variables $\theta_1$ and $\psi_2$ are
10 depicted in Fig. 1, for the reference system (reality) and a model version (model 0) for which the friction coefficient has been slightly modified.

These parameter changes induce slight modifications of the dynamics. In particular the system possesses two distinct weather regimes, depicted in Fig. 2(a) and (b): one characterised by a zonal circulation (see Fig. 2(c)), and another characterised by a blocking situation (see Fig. 2(d)). In the former case, the variables $\psi_2$ and $\psi_3$ characterising the strength of the meridional
15 anomalies are small, while in the latter case they are large, indicating indeed a blocking situation. This is different from

| Parameter description \ Experiment | Symbol \ System | Reality | Newtonian cooling modification | | Friction coefficient modification | |
|---|---|---|---|---|---|---|
| | | | Model 0 | Model 1 | Model 0 | Model 1 |
| Newtonian cooling coefficient | $h_d$ | 0.3 | 0.33 | 0.315 | 0.3 | |
| Atm. layers friction | $k_d$ | 0.1 | 0.1 | | 0.12 | 0.11 |
| Bottom layer friction | $k_d'$ | 0.01 | | | | |
| Domain aspect ratio | $n$ | 1.3 | | | | |
| Meridional temperature gradient | $\theta_1^\star$ | 0.2 | | | | |
| Mountain ridge altitude | $h_2$ | 0.4 | | | | |

**Table 1.** The main parameters used and modified in the experiments. Model 0 and model 1 are respectively the forecast model of the reality before and after the model change.

the situation considered in Reinhold and Pierrehumbert (1982), where two different blocking regimes coexist with the zonal regime.

## 4.1 Postprocessing experiments

The model described above with 10 modes ($n_\mathrm{a} = 10$) is used and two different postprocessing experiments are performed, one
involving the Newtonian cooling parameter $h_d$ and another involving the friction parameter $k_d$ between the two atmospheric layers. The parameter values detailed above correspond to the long-term reference (i.e. the reality). A first model is defined (model 0) which is a copy of the 2-layer quasi-geostrophic model defining the reality, but the parameters $h_d$ or $k_d$ are slightly changed, i.e. the model error of the forecasting system lies in either the Newtonian cooling or the friction parameter. Then, as in Section 3, a model change is imposed, leading to another forecasting model (model 1) that can either improve or degrade the model error by a factor $\kappa$. The parameter variations involved in these experiments are detailed in Table 1. Without loss of generality, we consider model changes that improve the representation of reality, in the sense that the amplitude of the model errors in model 1 is smaller than in model 0. The effect of the model change is depicted in Figs. 3 and 4 for the friction parameter experiment. These figures display the mean and the standard deviation of the model forecasts and observations coming from the reference forecasts, as a function of the lead time $\tau$. We have used a set of one million trajectories of each system to compute these averages.

In the framework of the EVMOS postprocessing scheme, the predictors and the predictands are the same nominal variable and no other predictors are used. In both experiments considered, the postprocessing parameters $\alpha$ and $\beta$ of the EVMOS for model 0, as well as $\hat{\alpha}$ and $\hat{\beta}$ for model 1, are computed. The main objective here is then to estimate the difference between the former and the latter using Ruelle response theory. The approach in a multivariate setting is presented below.

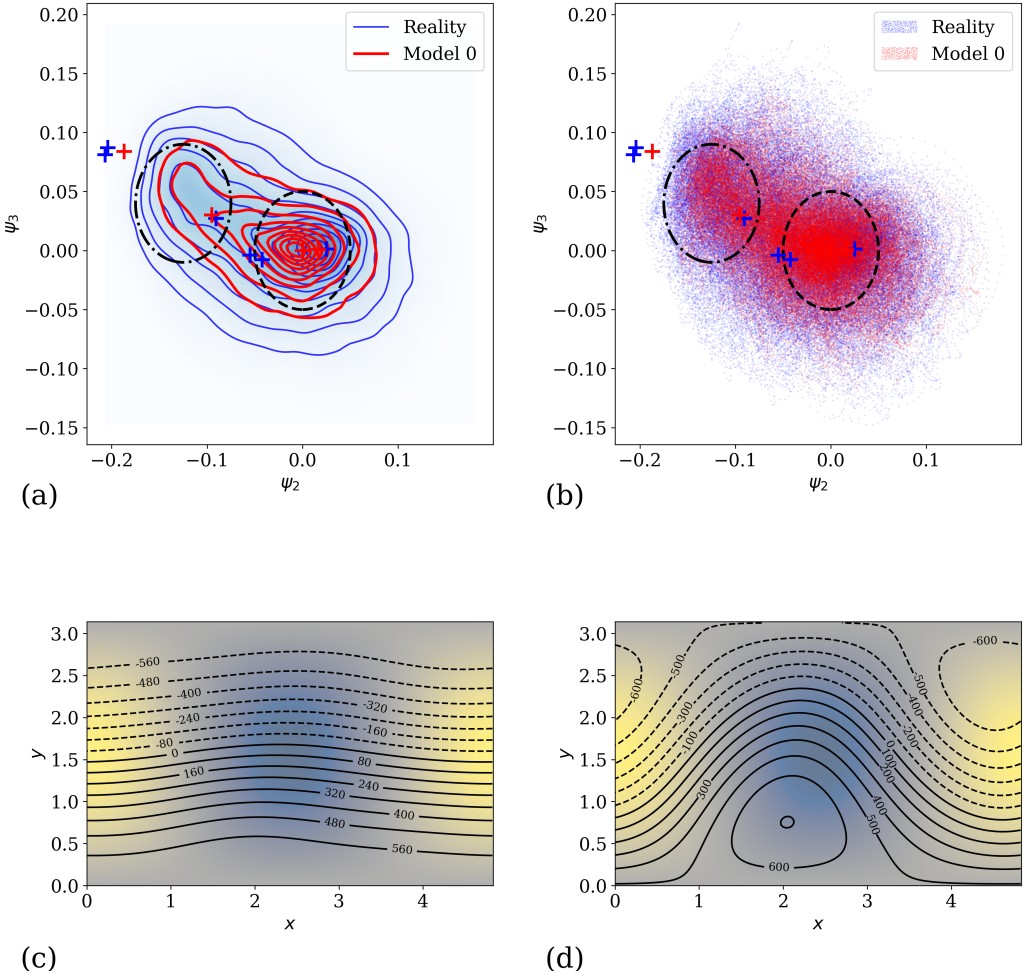

**Figure 2.** Attractors for the experiment with modification of the friction coefficient: (a) Two-dimensional isodensity of the attractors estimated with a Gaussian kernel density estimator for the variables $\psi_2$ and $\psi_3$, (b) two-dimensional scatter plot of the attractors for the variables $\psi_2$ and $\psi_3$. The attractors of the reality and model 0 are qualitatively similar, with two different parts which are indicated by ellipses. The blue and red crosses correspond respectively to equilibrium points of the reference model (the reality) and of the model 0, respectively. The dashed ellipse corresponds on average to a zonal circulation depicted on panel (c). The dashed-dotted ellipse corresponds on average to a blocking situation depicted in panel (d). In both panels (c) and (d), the underlying colour map denotes the orography on the domain, and the contours the geopotential height anomaly at 500 hPa.

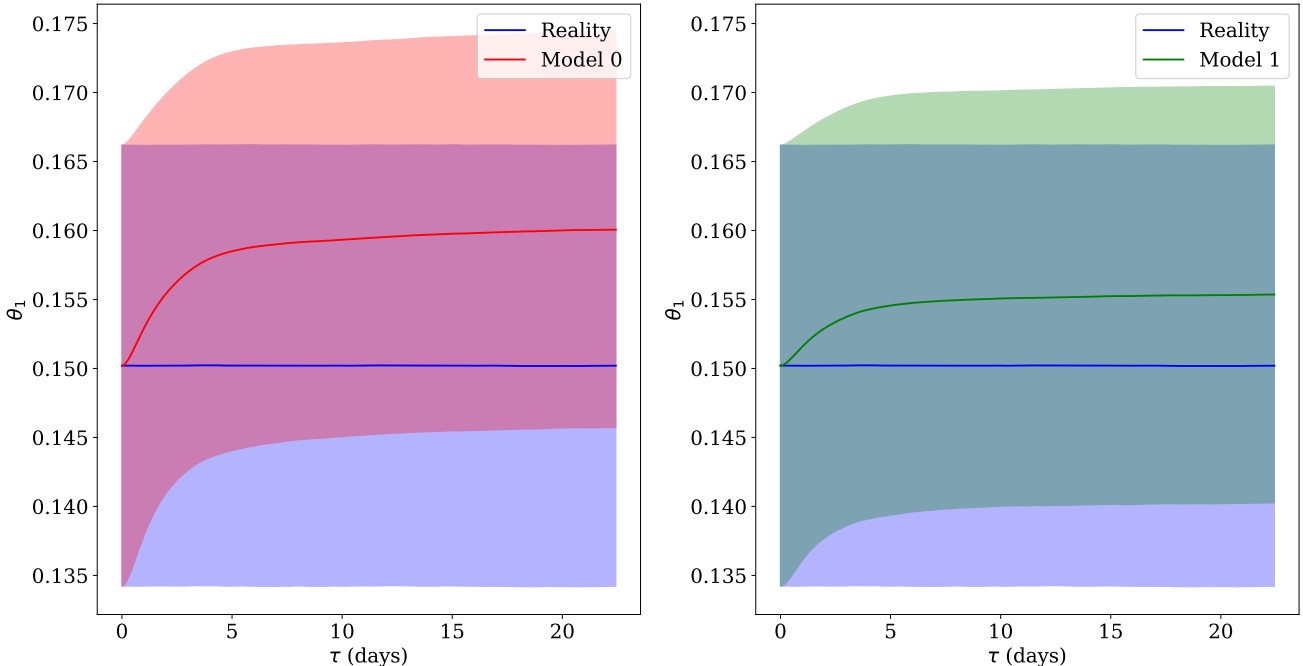

**Figure 3.** Behaviour of the averages as a function of the lead time $\tau$ in the reality and the forecast models before (left panel) and after (right panel) the model change, for the experiment with modification of the friction coefficient (see Table 1). The variable considered is the temperature meridional gradient $\theta_1$. The solid lines denote the mean while the shaded areas denote the one standard deviation interval.

### 4.2 Model change, Response theory and the Tangent Linear model

Let us consider again the response theory described in section 3.3, but in the general multivariate deterministic case described in section 2. In the postprocessing framework, models 0 and 1 evolve in time from a set of initial conditions taken outside of their respective attractors. Response formulae found in Ruelle's work have to be adapted to take this into account. One therefore has to consider the density of initial conditions as the measure. For a system with a time-independent perturbation $\boldsymbol{\Psi}(\hat{\boldsymbol{y}})$,

$$\dot{\hat{\boldsymbol{y}}} = \boldsymbol{F}(\hat{\boldsymbol{y}}) + \boldsymbol{\Psi}(\hat{\boldsymbol{y}}) = \hat{\boldsymbol{F}}(\hat{\boldsymbol{y}}), \tag{47}$$

an observable $A$ with average $\langle A(\tau)\rangle_{\boldsymbol{y}}$ at the lead time $\tau$ for the system

$$\dot{\boldsymbol{y}} = \boldsymbol{F}(\boldsymbol{y}) \tag{48}$$

has a first order response of

$$\delta\langle A(\tau)\rangle_{\boldsymbol{y}} = \int \mathrm{d}\boldsymbol{y}_0\, \rho_0(\boldsymbol{y}_0)\, \boldsymbol{\delta y}(\tau)^{\mathsf{T}} \cdot \boldsymbol{\nabla}_{\boldsymbol{f}^\tau(\boldsymbol{y}_0)}\, A \tag{49}$$

where $\boldsymbol{f}^\tau$ is the flow of the unperturbed system (48), $\rho_0$ is the distribution of initial conditions, and $\boldsymbol{\delta y}(\tau)$ is the solution of the equation $\dot{\boldsymbol{y}} + \dot{\boldsymbol{\delta y}} = \hat{\boldsymbol{F}}(\boldsymbol{y} + \boldsymbol{\delta y})$ which can be approximated at first order by the following linear inhomogeneous differential

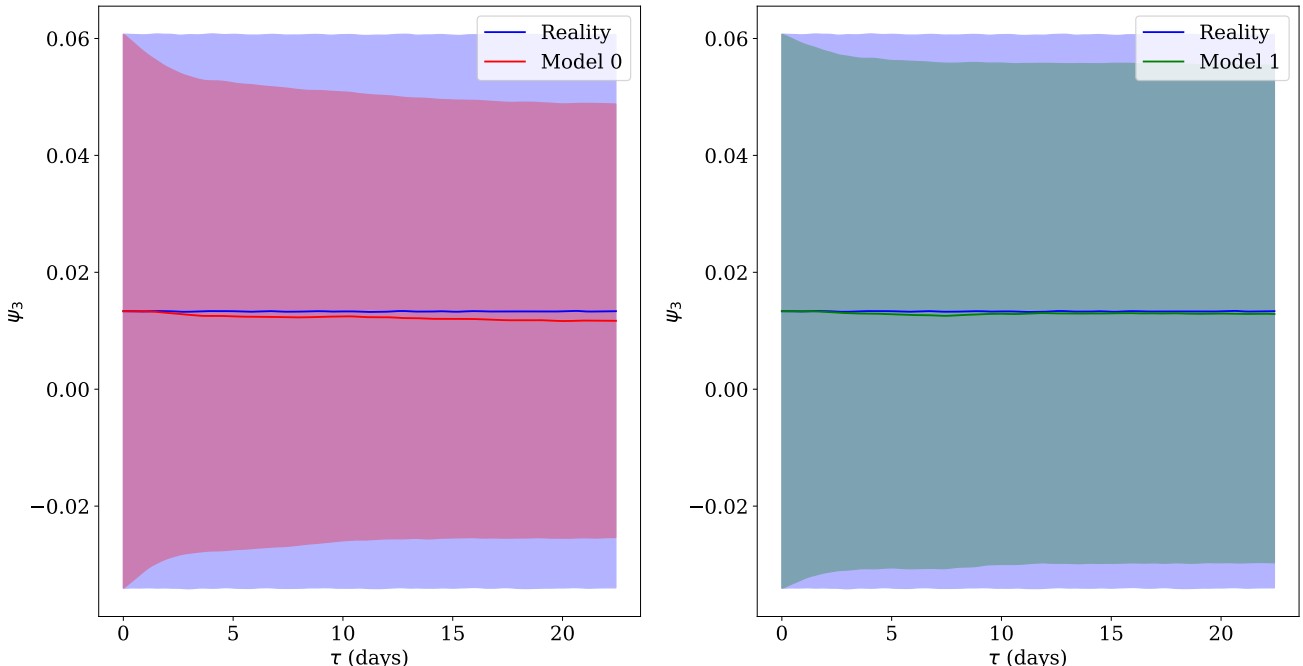

**Figure 4.** Same as Figure 3, but for the variable $\psi_3$ of the streamfunction $\psi$.

equation

$$\dot{\boldsymbol{\delta y}} = \boldsymbol{\nabla_y F} \cdot \boldsymbol{\delta y} + \boldsymbol{\Psi}(\boldsymbol{y}). \tag{50}$$

where $\boldsymbol{y}(\tau)$ is solution of the unperturbed equation (48) with initial condition $\boldsymbol{y}(0) = \boldsymbol{y}_0$ and we see that the systems (48) and (50) have to be integrated simultaneously (Gaspard, 2005). The homogeneous part of Eq. (50) is the well-known tangent linear model of the system and here it has to be solved with an additional boundary term which is the perturbation itself.

Equation (49) is derived in Appendix A, and can be computed in the same way as the averages depicted in Figs. 3 and 4, by averaging over multiple initial conditions of the reference system. Since we initialise the unperturbed (model 0) and perturbed systems (model 1) with the same initial conditions, the initial state of the tangent model (50) is $\boldsymbol{\delta y}(0) = 0$. Therefore we do not estimate the impact of the observation or assimilation errors, but rather the direct impact of the model errors viewed as time-independent perturbations. The formulation of the problem and Eq. (50) can be adapted to take these errors into account, as described for instance by Nicolis (2016).

In what follows, we will numerically integrate Eq. (50) to evaluate the response on the average due to the perturbation induced by the model change. This will in turn, as in Section 3, enables us to compute the postprocessing parameters for the new model.

## 4.3 Main results

For each of the two experiments detailed in Table 1, we start by obtaining one million observations of the reality that will be used to initialise the forecast models. For each observation, this is done by starting the model $x$ (the reference) with a random initial condition and running it for a very long time (100000 nondimensionalised time units) to achieve convergence to its global attractor. Once the observations have been obtained, we run the reference model, model 0 and model 1 over 200 time units (corresponding to roughly 22 days) to obtain the reality and the forecasts. The systems have been integrated using the fourth-order Runge-Kutta integration scheme with a time-step of 0.1 time unit corresponding to 16.15 minutes. The averaging over the one million trajectories of the reality and of the forecasts at each lead time is used to compute the postprocessing coefficients $\alpha$ and $\beta$ of the EVMOS by using formulas (17) and (18). For each predictand, the corresponding model variable is used as the unique predictor.

The response-theory approximations of the averages of the model $\hat{y}$ (model 1) averages are obtained by integrating the linearised equations of model 0 along its trajectories with the perturbation $\Psi$ as inhomogeneous term. This is done by integrating Eq. (50) over a lead time of 200 time units with a zero initial condition, using the same integration scheme as before. It gives us the integrand of Eq. (49) for each trajectory, and the integral is then approximated as the average of this integrand over the whole set of trajectories. The result of this integration and averaging is shown in Figs. 5 and 6 for the first and second moment of the variable $\theta_1$. The results for other variables are available in the supplementary material. The black curve shows the moments of model 0 with the addition of their linear response $\delta\langle\theta_1\rangle$ and $\delta\langle\theta_1^2\rangle$ to the perturbation $\Psi$. This curve agrees well with the green curves of the model 1 moments up to a lead time of 4-5 days, showing the efficiency of response theory. Note that in contrast with the calculation of the averages shown in Figs. 3 and 4 and computed with one million trajectories, we have here considered a limited subset of 10000 trajectories of model 0 and its tangent to compute the corrections to these averages. The correction of the moments of model 1 are accurate until 4 days for both experiments. After this critical lead time, obtaining a good accuracy requires a huge increase in the number of forecasts and tangent model integrations to perform the averaging. This problem is well-known (Nicolis, 2003; Eyink et al., 2004) and is due to the appearance of fat-tails in the distribution of the perturbations $\delta y$ in the integrand of Eq. (49). As it can be seen in Fig. 7 for the perturbations on $\theta_1$, the problem worsens with the increase of the lead time: initially the distributions are near-Gaussian and fat-tails appears progressively. Therefore, the number of samples of $\delta y$ needed to converge to the correct mean up to a certain precision increases exponentially as the lead time increases. This problem has consequences on the method used to perform the average. Indeed, to avoid rare and unrealistic extreme responses of the system located far in the tails of the distributions, outliers above a certain threshold (set to 3 nondimensional units) have been removed from the averaging.

The moments obtained by the response theory approach are used to compute new EVMOS postprocessing $\alpha$ and $\beta$ coefficients, thanks to the formulas (17) and (18). These corrected coefficients for variable $\theta_1$ are shown in Fig. 8 for the experiment with modification of the Newtonian cooling coefficient and in the panels (c) and (d) of Fig. 11 for the experiment with modification of the friction coefficient. In Figs. 9 and 10, we compare the performances of the four postprocessing schemes hence obtained: the postprocessing of model 0 (red curves) and 1 (green curves) obtained by averaging over their trajectories (fore-

casts), and the postprocessing of model 1 obtained with the past model 0 forecasts (green "+" crosses) and with the response theory approach (black "×" crosses). In the panel (a) of these figures, the mean square error (MSE) between the trajectories of the models and the reference trajectories is displayed by solid curves, while the MSE between both models correction and the reference is depicted by dash-dotted curves. The EVMOS postprocessing is able to partly correct the forecasts, reducing the

MSE until a lead time of the order of a few times the model's Lyapunov time (the inverse of the leading Lyapunov exponent). After that, the MSE curves of the postprocessed and uncorrected forecasts converge toward a plateau corresponding to twice the variance of the reference solution (Vannitsem, 2009). Here, the statistical postprocessing corrections are indeed efficient until lead times of 4-5 days, with a skill of the corrections decreasing with the lead time. Thus the EVMOS schemes become not better than the original models after roughly 4 days. Note also that even if the model change is small, the postprocessing

using the past forecasts of model 0 (green "+" crosses) completely fails to correct model 1 forecasts, highlighting the need for an adaptation of the postprocessing to the model change. In contrast, the adaptation with the response-theory method (black "×" crosses) produces valid corrections until 4 days ahead. In the panels (b) and (c) of Figs. 9 and 10, the mean and variance of the corrected forecasts are compared with those of the original models. Again, the corrections obtained with response theory are efficient until 4 days for the postprocessing schemes.

In conclusion, the correction of model 1 using the response-theory EVMOS matches almost perfectly the score of the "exact" EVMOS obtained with the forecasts of model 1 (dash-dotted green curve), up to a 4 days lead time. After that lead time, the errors due to the fat-tails in the response of the first moments of the statistics induce errors in the variance needed to compute the $\alpha$ and $\beta$ coefficients (see Eqs. (17) and (18)). These coefficients therefore degrade sharply after 4 days, as shown by the solid black curve in Fig. 8 and in Fig. 11(c) and (d). This in turn induces a degradation of the response theory postprocessing

scheme. Nevertheless, this limitation of response theory is not a concern here, since after a lead time of 4 days, the EVMOS skill improvement vanishes anyway.

## 5  Discussion and Conclusions

Statistical postprocessing techniques used to correct numerical weather predictions (NWP) require substantial past forecast and observation databases. In the case of a model change, which frequently occurs during the normal life cycle of an operational

forecast model, one has to reforecast the entire database of past forecasts (Hagedorn et al., 2008; Hamill et al., 2008) to update the postprocessing coefficients and parameters. In the present work, we proposed a new methodology based on response theory to produce these new coefficients without having to reforecast. Instead, the database of past forecasts is reused to perform integrations in the tangent space of the model. It allows to obtain the new postprocessing coefficients as modifications of the older ones. These new coefficients were shown to be accurate enough within the lead time range for which the postprocessing

corrections improve the forecast.

Figure 11 summarises the main results of this work, with the quasi-geostrophic system described in Section 4, but using a different number $m$ of trajectories of model 0 and its tangent model to compute the response-theory corrections. It shows

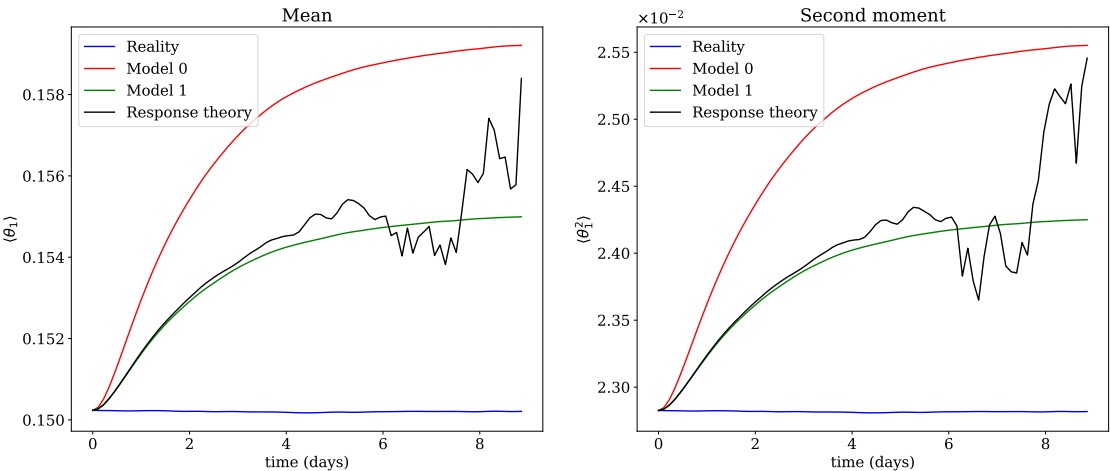

**Figure 5.** Corrections of the moments of $\theta_1$ from model 0 to model 1 using the response theory formula (49), for the experiment with modification of the friction coefficient.

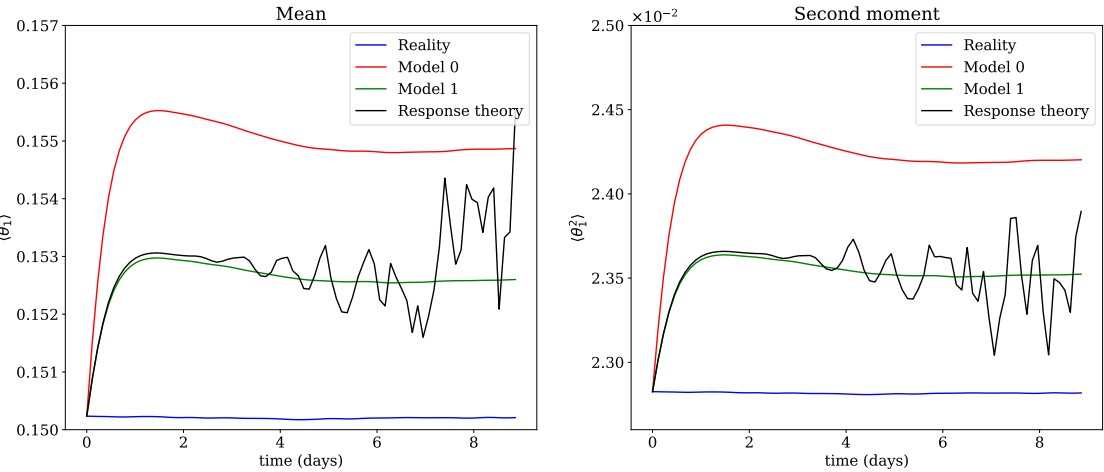

**Figure 6.** Corrections of the moments of $\theta_1$ from model 0 to model 1 using the response theory formula (49), for the experiment with modification of the Newtonian cooling coefficient.

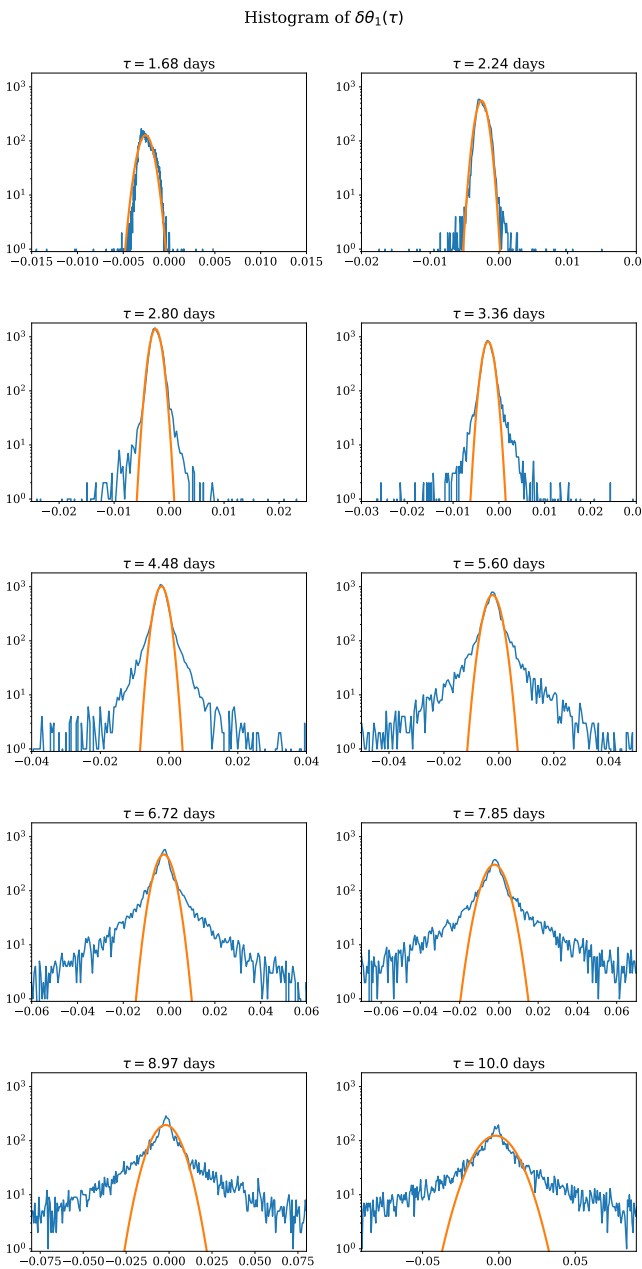

**Figure 7.** Histograms of the solutions $\delta\theta_1(\tau)$ of the equation (50) for the perturbation $\boldsymbol{\delta y}(\tau)$ (with $\theta_1 = y_{11}$) along the trajectories of model 0, for different values of the lead time $\tau$. The solid orange curves are fits of a Gaussian distribution function to the different histograms. The fat-tail phenomenon described in Eyink et al. (2004) is apparent and becomes more prominent as the lead time increases.

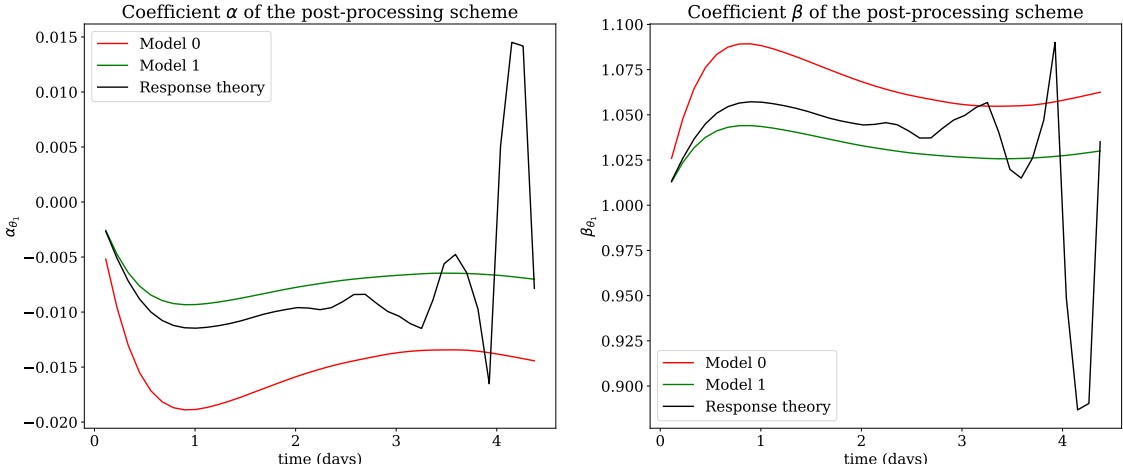

**Figure 8.** Coefficients $\alpha$ and $\beta$ of the postprocessing schemes of variable $\theta_1$ and their correction using the response theory, for the experiment with modification of the Newtonian cooling coefficient.

that up to a lead time of 2 days, good postprocessing scheme coefficients are obtained even with a mere 20 integrations in the tangent space.

Note however that in the context of this conceptual model, good estimates of the postprocessing coefficients $\alpha$ and $\beta$ can be obtained by simply using a small set of reforecasts. It is indeed enough to directly integrate the updated model 1, given by the
non-linear equation (47), with only 20 trajectories. So the response theory approach in the present case cannot really compete with the simple reforecasting method. How this can be improved in an operational context is an important question that should be addressed in the future. For instance, we can use a simplified tangent linear model to reduce the computational burden, as often used in data assimilation (Bonavita et al., 2017). This approach could also be implemented for short-range forecasts, say from 1 to 3 days.
The response-theory is efficient because the model changes are assumed to be small in comparison with the original parameterisation of the models. The method cannot improve a postprocessing scheme, but it can efficiently adapt it to a new model version. As such, the success of this method also depends on the quality of the past postprocessing scheme. There are situations where linear response theory is known to fail, but statistical tests which allow to identify its breakdown have been derived in Gottwald et al. (2016) and in Wormell and Gottwald (2018). In addition, the approach presented here applies only
for models for which a tangent model is available. The model change itself has to be provided as an analytic function, which can in some circumstances limit the applicability of the approach.

To test this approach, we have focused on the EVMOS statistical postprocessing method, but other methods could be considered as well. The only requirement is that the outcome of the minimisation of the cost function uses averages of the systems being considered. For instance, member-by-member methods that correct both the mean square errors and the spread of the

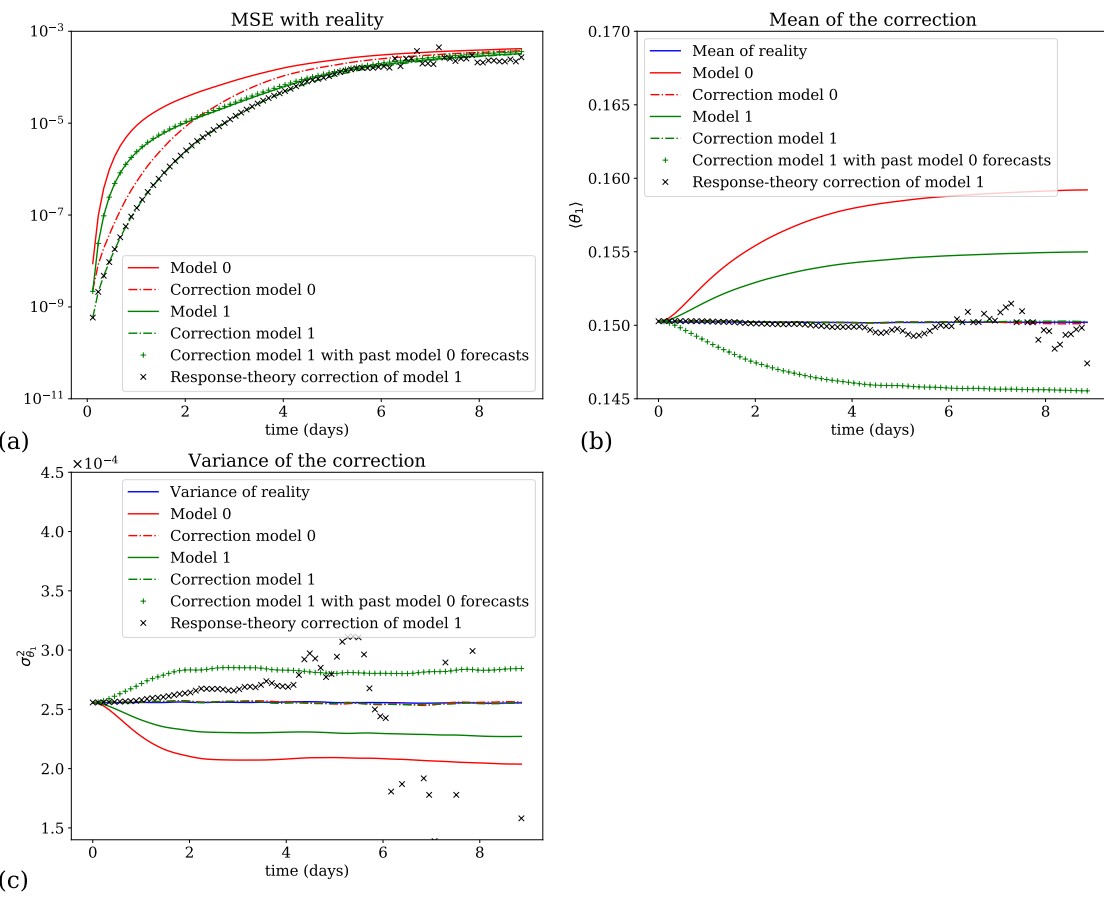

**Figure 9.** Performance of the corrections on the variable $\theta_1$ for the experiment with the modification of the friction coefficient. (a) Mean square error (MSE) evolution between the different forecasts and their correction, and the reality. (b) Mean of the different trajectories (reality, model 0 and 1) and corrected forecasts. (c) Variance of the different trajectories and corrected forecasts.

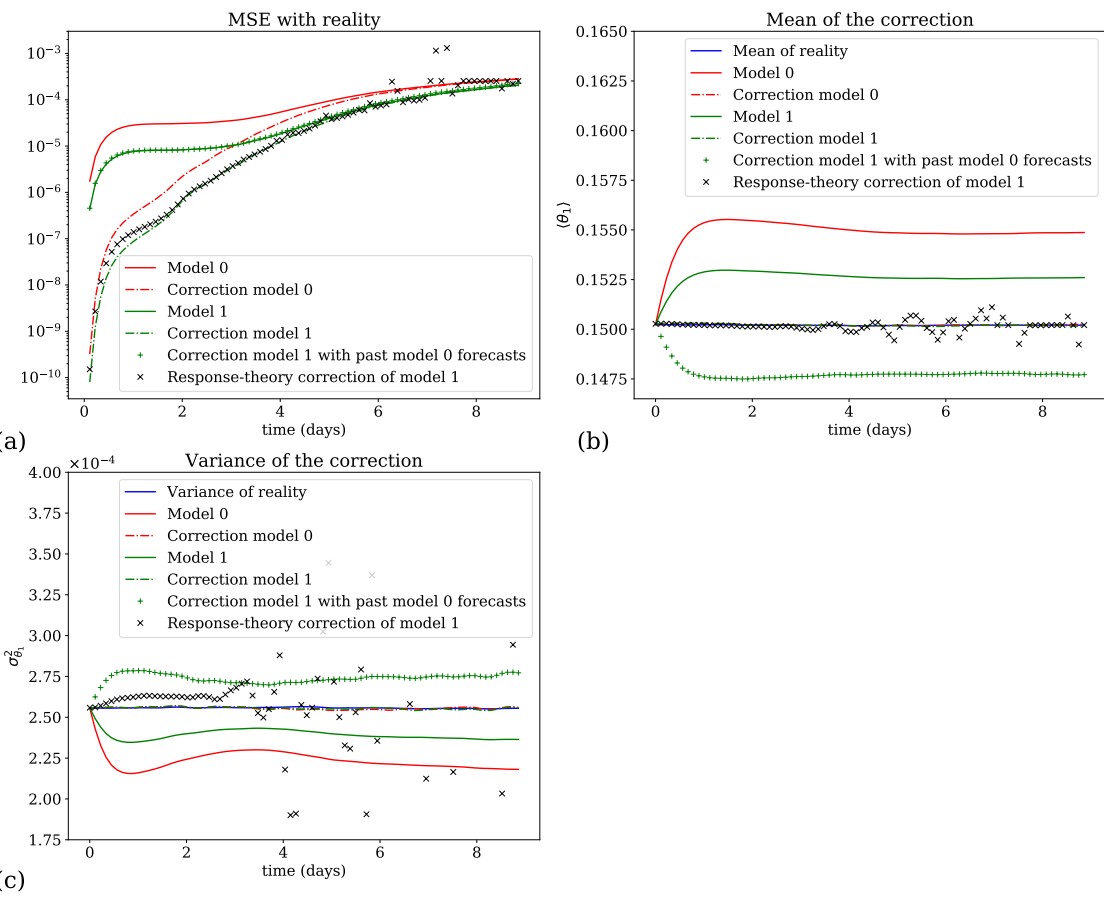

**Figure 10.** Performance of the corrections on the variable $\theta_1$ for the experiment with the modification of the Newtonian cooling coefficient. (a) Mean square error (MSE) evolution between the different forecasts and their correction, and the reality. (b) Mean of the different trajectories (reality, model 0 and 1) and corrected forecasts. (c) Variance of the different trajectories and corrected forecasts.

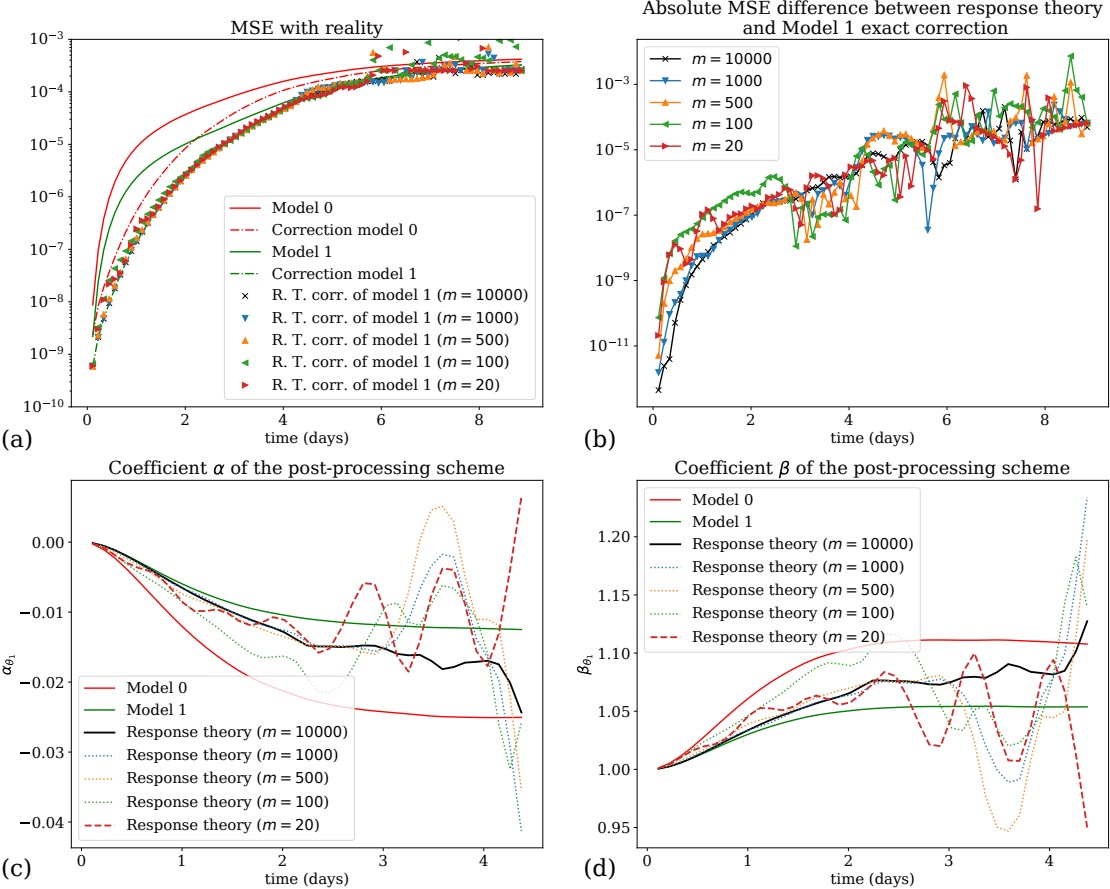

**Figure 11.** Comparison of the efficiency of the response theory correction for different numbers $m$ of trajectories used to average Eq. (49), for the experiment of varying the friction coefficient: (a) Mean square error with reality, (b) Absolute difference between the response theory correction and the correction based on the forecast of model 1, (c) and (d) Postprocessing coefficients $\alpha$ and $\beta$. On panels (b), (c) and (d), the higher (100000) and the lower (20) numbers are depicted respectively by a solid black line and a dashed red line. The other cases in-between are depicted by dotted lines.

ensemble while preserving the spatial correlation (Van Schaeybroeck and Vannitsem, 2015) could be considered. These methods generally use the covariance between the model forecasts and the observations as ingredient. Response theory can also be applied here since this covariance can be written as an average. This will be investigated in a future work, together with the applicability of the approach to parameters of probability distributions, as often used in meteorology (Vannitsem et al., 2018).

The impact of initial condition errors has not been addressed here, since the purpose was to demonstrate the applicability of the approach in a perfectly controlled environment. The main limiting issue of response theory in the present context is the presence of fat tails in the distribution of the perturbations $\delta y$ in the tangent model. This implies that beyond a certain lead time, typically 2-3 days for the synoptic scale, the number of trajectories of the tangent model needed for the averages to converge increases exponentially. This renders the approach impractical at lead times beyond 2-3 days. This is a well-known problem, which is typically due to the trajectories passing close to the stable manifolds structuring the dynamics of chaotic systems (Eyink et al., 2004), generating an extreme response of the system to the perturbations $\Psi$. This is possibly due to the exacerbated sensitivity of these manifolds to the perturbation of the system. We see two possibilities to overcome this issue in the case where a long lead-time correction is needed.

– First, as suggested by Eyink et al. (2004), the problem should be studied in other systems. It might be resolved by itself in other systems. Indeed, in very large atmospheric systems, the encounter of such manifolds might become more rare. This could be related to the chaotic hypothesis (Gallavotti and Cohen, 1995a, b) which states that large systems can be considered to behave like Axiom-a hyperbolic systems for the physical quantities of interest, and thus Ruelle response theory (Ruelle, 2009) might get better as the dimensionality of a system increases. This hypothesis would be interesting to test in current state-of-the-art NWP systems.

– Secondly, another avenue would be to adapt the techniques based on the Covariant Lyapunov vectors (CLVs) or on unstable periodic orbits (UPOs) to non-stationary dynamics. These techniques were recently introduced (Wang, 2013; Ni and Wang, 2017; Ni, 2019; Lasagna, 2019; Lasagna et al., 2019) to deal with stationary responses of chaotic systems, i.e. the response of a system that lies on its attractor.

The CLVs methods mentioned focus on finding an adjoint representation (Eyink et al., 2004) of the response, while in the present work the approach is based on forward integrations (direct method). The adjoint representation allows to change easily the perturbation function $\Psi$ for a fixed observable $A$, while the direct method enables to consider different observables while keeping the perturbation function fixed. The adjoint representation, however, requires one to integrate the tangent model backward in time. Therefore, its accuracy depends on the absolute value of the smallest Lyapunov exponent of the system, which might render its results less good than the direct forward representation.

In conclusion, the response-theory approach developed here is an effective method to deal with the problem of the impact of model change on the postprocessing scheme. Its main advantage is to be computed on the past model version and does not require reforecasts of the full model. Its operational implementation, however, is still an open question that should be addressed in the future.

*Code availability.* The quasi-geostrophic model used is called QGS and was obtained by adapting the Python code of the MAOOAM ocean-atmosphere model (De Cruz et al., 2016), following the model description in Cehelsky and Tung (1987). It was recently released on Zenodo (Demaeyer and De Cruz, 2020) and is also available at https://github.com/Climdyn/qgs. The additional notebooks computing the response to model changes and generating the figures are also provided as supplementary material. They have been released on Zenodo as well (Demaeyer, 2020), and are available at https://github.com/jodemaey/Postprocessing_and_response_theory_notebooks.

## Appendix A: Non-stationary response theory

We consider a perturbed autonomous dynamical system

$$\dot{\hat{\boldsymbol{y}}} = \boldsymbol{F}(\hat{\boldsymbol{y}}) + \boldsymbol{\Psi}(\hat{\boldsymbol{y}}) = \hat{\boldsymbol{F}}(\hat{\boldsymbol{y}}) \tag{A1}$$

with a prescribed distribution of initial conditions $\rho_0$. For the unperturbed system

$$\dot{\boldsymbol{y}} = \boldsymbol{F}(\boldsymbol{y}), \tag{A2}$$

an observable $A$ has the average at time $\tau$

$$\langle A(\tau) \rangle_{\boldsymbol{y}} = \int \mathrm{d}\boldsymbol{y}_0 \, \rho_0(\boldsymbol{y}_0) \, A(\boldsymbol{f}^\tau(\boldsymbol{y}_0)) = \int \mathrm{d}\boldsymbol{y} \, \rho_\tau(\boldsymbol{y}) \, A(\boldsymbol{y}) \tag{A3}$$

where $\boldsymbol{f}^\tau$ is the flow of the unperturbed system (A2) and where $\rho_\tau$ is the distribution obtained by propagating the initial distribution $\rho_0$ with the Liouville equation (Gaspard, 2005). In this section, the variation of this average due to the presence of the perturbation is evaluated,

$$\langle A(\tau) \rangle_{\hat{\boldsymbol{y}}} = \langle A(\tau) \rangle_{\boldsymbol{y}} + \delta \langle A(\tau) \rangle_{\boldsymbol{y}} + \delta^2 \langle A(\tau) \rangle_{\boldsymbol{y}} + \ldots \tag{A4}$$

In other words, we compute the average of $A$ in system (A1)

$$\langle A(\tau) \rangle_{\hat{\boldsymbol{y}}} = \int \mathrm{d}\boldsymbol{y}_0 \, \rho_0(\boldsymbol{y}_0) \, A(\hat{\boldsymbol{f}}^\tau(\boldsymbol{y}_0)) \tag{A5}$$

as a perturbation of the average (A3) in the unperturbed system (A2). Here, $\hat{\boldsymbol{f}}^\tau$ is the flow of the perturbed system (A1). In the following, we will derive these corrections thanks to a Kubo-type perturbative expansion (Lucarini, 2008) that amounts to constructing a Dyson series in the interaction picture framework where the perturbation is seen as an interaction Hamiltonian (Wouters and Lucarini, 2012). We start by considering the time evolution of the observable $A$ in the system (A1):

$$\frac{\mathrm{d}}{\mathrm{d}\tau} A\left(\hat{\boldsymbol{f}}^\tau(\boldsymbol{y}_0)\right) = (\mathcal{L}_0 + \mathcal{L}_1) \, A\left(\hat{\boldsymbol{f}}^\tau(\boldsymbol{y}_0)\right) \tag{A6}$$

with the operators

$$\begin{cases} \mathcal{L}_0 \, A(\boldsymbol{y}) &= \boldsymbol{F}(\boldsymbol{y})^\mathsf{T} \cdot \boldsymbol{\nabla}_{\boldsymbol{y}} A \\ \mathcal{L}_1 \, A(\boldsymbol{y}) &= \boldsymbol{\Psi}(\boldsymbol{y})^\mathsf{T} \cdot \boldsymbol{\nabla}_{\boldsymbol{y}} A \end{cases} \tag{A7}$$

and define an interaction observable as

$$A_I(\tau, \boldsymbol{y}_0) = \Pi_0(-\tau) A\left(\hat{\boldsymbol{f}}^\tau(\boldsymbol{y}_0)\right) \tag{A8}$$

with $\Pi_0(\tau) = \exp(\mathcal{L}_0 \tau)$. It is easy to show that the interaction observable satisfies the differential equation:

$$\frac{\mathrm{d}}{\mathrm{d}\tau} A_I(\tau, \boldsymbol{y}_0) = \mathcal{L}_I(\tau) A_I(\tau, \boldsymbol{y}_0) \tag{A9}$$

with the interaction operator $\mathcal{L}_I(\tau) = \Pi_0(-\tau)\mathcal{L}_1\Pi_0(\tau)$. The solution to this equation is

$$A_I(\tau, \boldsymbol{y}_0) = A_I(0, \boldsymbol{y}_0) + \int_0^\tau \mathrm{d}s_1\, \mathcal{L}_I(s_1) A_I(s_1, \boldsymbol{y}_0) = A(\boldsymbol{y}_0) + \int_0^\tau \mathrm{d}s_1\, \mathcal{L}_I(s_1) A_I(s_1, \boldsymbol{y}_0) \tag{A10}$$

which can be rewritten as

$$A\left(\hat{\boldsymbol{f}}^\tau(\boldsymbol{y}_0)\right) = \Pi_0(\tau)A(\boldsymbol{y}_0) + \int_0^\tau \mathrm{d}s_1\, \Pi_0(\tau - s_1)\mathcal{L}_1\Pi_0(s_1) A_I(s_1, \boldsymbol{y}_0). \tag{A11}$$

Iteratively replacing the interaction observable by the formula (A10) finally leads to the Dyson series:

$$
\begin{aligned}
A\left(\hat{\boldsymbol{f}}^\tau(\boldsymbol{y}_0)\right) = \ & \Pi_0(\tau)A(\boldsymbol{y}_0) + \int_0^\tau \mathrm{d}s_1\, \Pi_0(\tau - s_1)\mathcal{L}_1\Pi_0(s_1) A(\boldsymbol{y}_0) \\
& + \int_0^\tau \mathrm{d}s_1 \int_0^{s_1} \mathrm{d}s_2\, \Pi_0(\tau - s_1)\mathcal{L}_1\Pi_0(s_1 - s_2)\mathcal{L}_1\Pi_0(s_2) A(\boldsymbol{y}_0) + \dots
\end{aligned}
\tag{A12}
$$

Using the definitions (A3) and (A5), as well as the fact that

$$g\left(\boldsymbol{f}^\tau(\boldsymbol{y}_0)\right) = \Pi_0(\tau)g(\boldsymbol{y}_0) \tag{A13}$$

for any smooth function $g$, we get finally a formula for the perturbations in Eq. (A4):

$$\langle A(\tau)\rangle_{\hat{\boldsymbol{y}}} = \langle A(\tau)\rangle_{\boldsymbol{y}} + \int_0^\tau \mathrm{d}s_1 \int \mathrm{d}\boldsymbol{y}_0\, \rho_0(\boldsymbol{y}_0)\Pi_0(\tau - s_1)\mathcal{L}_1\Pi_0(s_1) A(\boldsymbol{y}_0) + \dots \tag{A14}$$

We will now focus on the first term of this expansion, but the subsequent orders of the response can be treated alike. We thus have

$$\delta\langle A(\tau)\rangle_{\boldsymbol{y}} = \int_0^\tau \mathrm{d}s_1 \int \mathrm{d}\boldsymbol{y}_0\, \rho_0(\boldsymbol{y}_0)\, \Psi\left(\boldsymbol{f}^{\tau - s_1}(\boldsymbol{y}_0)\right)^\mathsf{T} \cdot \boldsymbol{\nabla}_{\boldsymbol{f}^{\tau - s_1}(\boldsymbol{y}_0)} A\left(\boldsymbol{f}^\tau(\boldsymbol{y}_0)\right) \tag{A15}$$

which with the change of variable $s_1 \to t - \tau'$ can be rewritten as

$$\delta\langle A(\tau)\rangle_{\boldsymbol{y}} = \int_0^\tau \mathrm{d}\tau' \int \mathrm{d}\boldsymbol{y}_0\, \rho_0(\boldsymbol{y}_0)\, \Psi\left(\boldsymbol{f}^{\tau'}(\boldsymbol{y}_0)\right)^\mathsf{T} \cdot \boldsymbol{\nabla}_{\boldsymbol{f}^{\tau'}(\boldsymbol{y}_0)} A\left(\boldsymbol{f}^\tau(\boldsymbol{y}_0)\right) \tag{A16}$$

and then

$$\delta\langle A(\tau)\rangle_{\boldsymbol{y}} = \int_0^\tau \mathrm{d}\tau' \int \mathrm{d}\boldsymbol{y}_0\,\rho_0(\boldsymbol{y}_0)\,\boldsymbol{\Psi}\left(\boldsymbol{f}^{\tau'}(\boldsymbol{y}_0)\right)^{\mathsf{T}} \cdot \left(\frac{\partial \boldsymbol{f}^\tau(\boldsymbol{y}_0)}{\partial \boldsymbol{f}^{\tau'}(\boldsymbol{y}_0)}\right)^{\mathsf{T}} \cdot \boldsymbol{\nabla}_{\boldsymbol{f}^\tau(\boldsymbol{y}_0)} A \tag{A17}$$

$$= \int_0^\tau \mathrm{d}\tau' \int \mathrm{d}\boldsymbol{y}_0\,\rho_0(\boldsymbol{y}_0)\,\boldsymbol{\Psi}\left(\boldsymbol{f}^{\tau'}(\boldsymbol{y}_0)\right)^{\mathsf{T}} \cdot \mathbf{M}\left(\tau - \tau', \boldsymbol{f}^{\tau'}(\boldsymbol{y}_0)\right)^{\mathsf{T}} \cdot \boldsymbol{\nabla}_{\boldsymbol{f}^\tau(\boldsymbol{y}_0)} A \tag{A18}$$

$\mathbf{M}$ is the fundamental matrix (Gaspard, 2005; Nicolis, 2016) of the homogeneous part of the linear differential equation

$$\dot{\boldsymbol{\delta y}} = \boldsymbol{\nabla}_{\boldsymbol{y}} \boldsymbol{F} \cdot \boldsymbol{\delta y} + \boldsymbol{\Psi}(\boldsymbol{y}) \tag{A19}$$

where $\boldsymbol{y}$ is solution of Eq. (A2) with initial condition $\boldsymbol{y_0}$, and we have the definition

$$\mathbf{M}(t, \boldsymbol{y}) = \frac{\partial \boldsymbol{f}^t(\boldsymbol{y})}{\partial \boldsymbol{y}}. \tag{A20}$$

Equation (A19) is the linearised approximation of equation (A1):

$$\dot{\boldsymbol{y}} + \dot{\boldsymbol{\delta y}} = \boldsymbol{F}(\boldsymbol{y} + \boldsymbol{\delta y}) + \boldsymbol{\Psi}(\boldsymbol{y} + \boldsymbol{\delta y}) \tag{A21}$$

that provides a tool to estimate Eq. (A18). Indeed, since the solution of Eq. (A19) can be written as

$$\boldsymbol{\delta y}(\tau) = \int_0^\tau \mathrm{d}\tau'\,\mathbf{M}\left(\tau - \tau', \boldsymbol{f}^{\tau'}(\boldsymbol{y_0})\right) \cdot \boldsymbol{\Psi}\left(\boldsymbol{f}^{\tau'}(\boldsymbol{y_0})\right), \tag{A22}$$

we can write the first order variation of the average of the observable $A$ in term of these solutions:

$$\delta\langle A(\tau)\rangle_{\boldsymbol{y}} = \int \mathrm{d}\boldsymbol{y}_0\,\rho_0(\boldsymbol{y}_0)\,\boldsymbol{\delta y}(\tau)^{\mathsf{T}} \cdot \boldsymbol{\nabla}_{\boldsymbol{f}^\tau(\boldsymbol{y}_0)} A \tag{A23}$$

The interpretation of this equation is that the averaging of an observable over the trajectories of the linear approximation (A21)

of the perturbation equation (A1) provides the first order response of the observable. It is the main ingredient used to compute the new postprocessing scheme in the present work. It is explained in detail in Sections 3.3 and 4.2.

## Appendix B:  The quasi-geostrophic model equations

The ordinary differential equations of the model are given by

$$
\dot{\psi}_i = -a_{i,i}^{-1} \sum_{j,m=1}^{n_a} b_{i,j,m} \left( \psi_j \psi_m + \theta_j \theta_m \right) - \frac{a_{i,i}^{-1}}{2} \sum_{j,m=1}^{n_a} g_{i,j,m} \, h_m \left( \psi_j - \theta_j \right)
$$

$$
- \beta \, a_{i,i}^{-1} \sum_{j=1}^{n_a} c_{i,j} \, \psi_j - \frac{k_d}{2} \left( \psi_i - \theta_i \right) \tag{B1}
$$

$$
\dot{\theta}_i = -a_{i,i}^{-1} \sum_{j,m=1}^{n_a} b_{i,j,m} \left( \psi_j \theta_m + \theta_j \psi_m \right) + \frac{a_{i,i}^{-1}}{2} \sum_{j,m=1}^{n_a} g_{i,j,m} \, h_m \left( \psi_j - \theta_j \right)
$$

$$
- \beta \, a_{i,i}^{-1} \sum_{j=1}^{n_a} c_{i,j} \, \theta_j + \frac{k_d}{2} \left( \psi_i - \theta_i \right) - 2 \, k_d' \, \theta_i + a_{i,i}^{-1} \, \omega_i \tag{B2}
$$

$$
\dot{\theta}_i = - \sum_{j,m=1}^{n_a} g_{i,j,m} \, \psi_j \theta_m + \frac{\sigma}{2} \, \omega_i + h_d \left( \theta_i^* - \theta_i \right) \tag{B3}
$$

where nondimensional parameters values and description can be found in Table 1 and section 4. $\beta$ is the meridional gradient of Coriolis parameter which has the nondimensional value 0.21 at 50 degrees of latitude (Reinhold and Pierrehumbert, 1982; Cehelsky and Tung, 1987). The vertical velocity $\omega_i$ can be eliminated, leading to equations (B2) and (B3) being reduced to a single equation for $\theta_i$. The parameter $\sigma$ is the nondimensional static stability of the atmosphere set typically to 0.2. The coefficients $a_{i,j}$, $g_{i,j,m}$, $b_{i,j,m}$ and $c_{i,j}$ are the inner products of the Fourier modes $F_i$ defined in section 4:

$$
a_{i,j} = \frac{n}{2\pi^2} \int_0^\pi \int_0^{2\pi/n} F_i(x,y) \, \nabla^2 F_j(x,y) \, \mathrm{d}x \, \mathrm{d}y = -\delta_{ij} \, a_i^2 \tag{B4}
$$

$$
g_{i,j,m} = \frac{n}{2\pi^2} \int_0^\pi \int_0^{2\pi/n} F_i(x,y) \, J \left( F_j(x,y), F_m(x,y) \right) \, \mathrm{d}x \, \mathrm{d}y \tag{B5}
$$

$$
b_{i,j,m} = \frac{n}{2\pi^2} \int_0^\pi \int_0^{2\pi/n} F_i(x,y) \, J \left( F_j(x,y), \nabla^2 F_m(x,y) \right) \, \mathrm{d}x \, \mathrm{d}y \tag{B6}
$$

$$
c_{i,j} = \frac{n}{2\pi^2} \int_0^\pi \int_0^{2\pi/n} F_i(x,y) \, \frac{\partial}{\partial x} F_j(x,y) \, \mathrm{d}x \, \mathrm{d}y \tag{B7}
$$

where the coefficients $a_i$ are given by Eq. (41) and where $J$ is the Jacobian present in the advection terms:

$$
J(S,G) = \frac{\partial S}{\partial x} \frac{\partial G}{\partial y} - \frac{\partial S}{\partial y} \frac{\partial G}{\partial x}. \tag{B8}
$$

*Competing interests.* Stéphane Vannitsem is a member of the editorial board of the journal. Jonathan Demaeyer declares that he has no conflict of interest.

*Acknowledgements.* This work is supported in part by the Postprocessing module of the NWP Cooperation program of the European Meteorological Network (EUMETNET). The authors warmly thank Lesley De Cruz for her suggestions throughout the manuscript. They also thank Michaël Zamo and the anonymous reviewer for their comments and suggested improvements.

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
