# Peer review of "Correcting for Model Changes in Statistical Postprocessing – An approach based on Response Theory"

_Nonlinear Processes in Geophysics, 2019_

## Referee Comment (RC1) · Michael Zamo (Referee) · 20 Dec 2019

[10pt,twoside]report fancyhdr,color,amsmath,amssymb  [LO,RE]Review of a Manuscript for Nonlinear Processes in Geophysics (NPG)

Correcting for Model Changes in Statistical Post-Processing - An approach based on Response Theory

**General comments**

The authors present a new approach to tackle an important problem in the field of statistical post-processing of numerical weather prediction (NWP) model: how to post-process a NWP model when it changed and an old post-processing method has been gitted but may be no more efficient on the new version of the NWP model? Indeed

post-processing usually requires a long archive of past forecasts and associated observations to efficiently learn the error structures of the NWP model with statistical methods.

Several approaches already exist to handle this problem:

- producing hindcasts with the new version of the NWP model. This is the ideal approach but is very resource intensive.

- using transfer learning approach. To the best of my knowledge, this approach has not been deeply studied for meteorology.

- using online post-processing techniques with short training windows. This allows to quickly update the post-processing, but can only correct according to the near past (which may be useless for rare events)

- using filtering techniques to correct in real-time the residual biases in the post-processed forecasts. This has the same limitations has online post-processing.

- aggregating several NWP models and post-processed version thereof. This may improve performance and allow to discard useless models, but requires to have available several (post-processed or raw) forecasts with different error structures.

The new proposed approach is based on response theory. It consists in correcting the regression coefficients of the (linear) post-processing model, based on information learned on a period were the old and new NWP models are available. This approach supposes that the changes in the NWP model are small.

The presentation is clear and the general idea well exposed. The presentation of the technical requirements is not very detailed for someone not working in modelisation, but is sufficient to understand the proposed process. The illustration with two toy models helps to clarify the approach and is a good proof-of-concept. However, it would be interesting to have a more realistic application of the approach on real data.
The problem this approach tries to solve is an important one for operational purposes. Another tool to solve it is therefore welcome and its efficiency and usefulness should be further assessed by the community of meteorologists. This article is thus a precious contribution to the post-processing litterature.

Naturally, since it is a brand new approach to an old problem, several questions arise that should, at the very least, be raised with, ideally, some proposals:

- since this approach supposes the changes in the model are small, we can wonder if it will really be useful for operations, where small changes in the model may not really impair post-processing. Furthermore, it should be interesting to see whether this approach improves over the use of dynamic filtering of residual biases (via Kalman filtering, classicaly). Do you have any hint about this alternative?

- the stated conditions for using this correcting seem very strong: a tangent model must be available, the model change has to be provided as an analytic function. These two conditions may not be observed or the models themselves may not be available. Would it be possible to follow the same procedure in a data-based approach? In other words, could it be possible to deduce the necessary correction if one has only the two sets of forecasts on a common period without access to the models themselves?

- the extend of post-processing methods that may be corrected in this way is not very clear. You state that *The only requirement is that the outcome of the minimisation of the cost functions uses averages of the systems being considered.* Does this mean that post-processing methods that do not use cost functions (such as random forest) are not eligible to this approach?

**Specific comments**

1. *page 3, line 5:* The response theory approach provides an efficient correction of the post-processing scheme up to a lead time of 3 days, which matches the lead-time window where the scheme's correction is efficient.
   Do you have a reference to support this claim that post-processing is useful only up to a lead time of 3 days?

2. *page 4, line 21:* More sophisticated approaches can be evaluated in the future.
   Could you develop about what kind of sophistication you are thinking about?

3. *page 8, line 2:* A 2-layer quasi-geostrophic atmospheric system on a $\beta$-plane with an orography is considered
   For the article to be self-contained, may you add a more comprehensive definition (with equations) for this system, maybe in the appendix?

4. *page 9, line 14:* This is different from the case considered in Reinhold and Pierrehumbert (1982), where two blocking regimes coexist with the zonal regime.
   Is there a simple explanation why your system has only one blocking regime instead of two?

5. *page 10, line :* Fig2 (a) and (b)
   These pictures are not very clear. In panel (b) only model 0 seems to have two attractors but from the text (page9, line 12 and elsewhere) I understand that reality also should have two attractors. Please can you clarify either the text or the pictures?

6. *page 14, line 18:* After this critical lead time, obtaining a good accuracy requires a huge increase in the number of forecasts and tangent model integrations to perform the averaging. This problem is well-known (Nicolis, 2003; Eyink et al., 2004) and is due to the appearance of fat-tails in the distribution of the perturbations $\delta y$ in the integrand of Eq. (39).
   Later on, you say that the distribution of perturbation has been approximated with
a gaussian distribution. In the conclusion, you propose to use the CLV method to get better corrections at farther lead times. I was wondering if we could improve the correction at long lead times by using a different distribution (with fatter tails) to approximate the perturbation distribution?

7. Based on the results, it seems this approach may require a very short period where both models are available. The length of the comon period (a few months?) seems to correspond to what may be available operationnaly in national weather services. This is a very good point, to check on real data, maybe in a future study.

**Technical corrections**

1. *page 1, line 2:* reforcasting
   Please change into *reforecasting*

2. *page 4, line 7:* given K past forecasts $y_k$
   Since $K$ is used for constants in the Ornstein-Uhlenbeck process, I would suggest using a different notation, such as $N$ for the number of past forecasts and $n$ for the forecast index.

3. *page 4, line 10:* $x_C(\tau)$
   I would suggest using $y_C(\tau)$ for the corrected forecast.

4. *page 9, line 13:* one characterised by a zonal circulation (see Fig. 1(c)), and another characterised by a blocking situation (see Fig. 1(d)).
   I guess the references are wrong and should read Fig. 2(c) and Fig. 2(d).

5. *page 19, line 4:* the minimisation of the cost functions uses averages
   Please change into *the minimisation of the cost function uses averages*

Michaël Zamo, Météo-France.

---

## Referee Comment (RC2) · Anonymous Referee #2 · 8 Jan 2020

**General Comments:**

The discussion paper presents a method for adapting a post-processing approach after a (small) change in the forecast model, as is often the case in operational weather forecasting. The innovative idea of this paper is to base the approach on response theory. Using a tangential linear model, transient response theory represents a correction method of a post-processing approach as long as the changes in the forecast model are small. Since the consideration of small changes in the prediction model is of great importance, the paper represents a potentially valuable contribution. I particularly congratulate the authors for the courage to base their approach on physical theory, in this

case transient response theory.

The general structure of the paper is clear and the paper is generally well written. The paper contains quite some mathematical derivations in the appendix and in the supplementary material. As such, it is a challange to understand the theory and may discurrage interested readers. The impact of the paper may be largely increase with some additional – simplified – explanation. I have added some ideas in this respect in the specific comments. Further, the description of the figure is held very short. I prefer to be guide through the figures, instead of just been given the main conclusion.

A carefull proofreading should be performed in terms of language and formulas (see a non-exhaustive list in the technical corrections).

**Specific Comments:**

The information in Eq. (13)-(14) and (16)-(17) is very redundant, since one only has to exchange the parameters. I suggest to compress this a little bit.

The application of the concept to the OU-processes is quite helpful. I would suggest to first formulate the response theory in general, and then in a separate section apply and explain the formulas for the OU case, followed by the post-processing. It would further be helpful to maybe see a figure of $\alpha(\tau)$ and $\beta(\tau)$.

With respect to the QG model it would be helpful to, on the one hand, see the model equations probably in the Fourier space, but, on the other hand, simplify the Fourier expansion (e.g. using complex expansion), since this is a quite standard Galerkin approach (you have to mention the boundary conditions).

The application of the response theory here is very not very clear to me, and I still do not quite see, where and how the tangent model comes into play.

Regarding Fig. 2 (b) you may guide the reader a little bit, e.g., mention, that $\psi_2$ and $\psi_3$ represent the strength of the zonal wavenumber 1 anomalies, which are both small in the dashed line ellipse, and large in the dashed-dot ellipse, which also seems to have

a strong zonal mean $\psi_1$ component. What is the relevance of the equilibrium ploints?

**Technical Corrections:**

page 1, line 2: ". . . cope with slight model change" -> ". . . cope with slight model changes"

p. 1, l. 3: "The response theory allows then to ..." -> Response theory allows us to ...". The usage of "allow" throughout the paper is incorrect. Please check with native speaker.

p. 1, l. 5: "averages involved": I only understood after reading the paper, what is meant by this.

p. 1, l 7: ". . . the application to the latter provide a proof-of-concept of the potential performance of response theory ..." -> ". . . the application to the latter provides a proof-of-concept and assesses the potential performance of response theory ..."

p. 1, l. 9: What is "in a more operational environment"? Maybe "in an environment more similar to the operational environment"

p. 2, l. 15: "These operational models, ...": Remove "these", since reference is not clear.

p. 2, l. 16: ". . . behaviours" -> ". . . behaviour" or maybe just ". . . processes".

p. 2, l. 17: "past forecasts" -> "forecasts in the past"

p. 2, l. 22ff: "Recent research have studied non-homogeneous regression with time-adaptative training scheme, where a trade-off have to be considered between large training data sets for stable estimation and the benefit of shorter training periods to adjust more rapidly to changes in the data (Lang et al., 2019). ": Language editing needed, e.g. "Recent research has investigated non-homogeneous regression with a time-adaptive training scheme, where a trade-off between a large training data set for stable estimates and the benefit of a shorter training period for faster adaptation to data
changes must be considered. (Lang et al., 2019). "

p. 2, l. 31: ". . . parameters variations as well as new terms in the tendencies are potential model change." -> ". . . parameter variations as well as new terms in the tendencies are potential model changes."

p. 3, l. 17: square bracket "< >" are not explained here, only later on. In general I find it confusing to refer here to averages, as these are to my understanding expectation values (which are indeed estimated via ensemble averages).

p. 3, eqs. (1), (2), (3): add $(\tau)$ to $\dot{x}$, $\dot{y}$, . . .

Eq. (5): remove $\cdot$, since this is not a scalar product.

Eq. (6): the numerator should be with a quare (I guess you're doing least quares).

p. 3, l. 29: "Note that if $\kappa = 1$, the correction is perfect.": Not absolutely true, since forecast and model still differ in $\lambda$.

p. 4, l. 20: "for correct ..." -> "for a correct ..."

p. 4, l. 25: ". . . of infinite number of ..." -> ". . . of an infinite number of ..."

Eq. (20): Remove ˆ from $\beta$ in denominator.

p. 6, l. 8: ". . . over model the $y$ forecasts" -> ". . . over the model $y$ forecasts"

p. 6, l. 19: "It is also assumed that there is no interference due to initial condition errors in the problem.": Please clarify, what you mean by this.

Eq. (23): $f^\tau()$ is only given later and specific for the OU process. Please state here, what this is.

Also Eq. (23): The term in the square brackes may be a scalar product in general, but as you have formulated $\psi_y$, it is not. Later on you use bold letters for vectors, right?

p. 7, l. 1: "its stochastic integrals ...">  This may need some explanation.

Eq. (28): Check formula: Isn't there missing $A(f^{\tau'}(x_o))$ and $A(f^{\tau}(x_o))$ should be $A(f^{\tau''}(x_o))$? Again is this really a scalar product?

p. 7, l. 23ff: You put $\sigma_y^2$ into square brackets, although these are already expected values, so the brackets make no sense here.

p. 8, l. 8+9: I assume, that the $n$ in the sine and cosine should not be there.

Section 3: The notation in Eqs. (31)-(33) and (34) may be misleading, since $x$ is a coordinate, while $\mathbf{x}$ is the vector of the coefficients of the spectral expansion. Further, $\mathbf{F}$ is not given.

p. 9, l. 13,14: Should be Fig. 2(c) and Fig. 2 (d).

p. 12, Eq. (39): Use either scalar product or transpose $^T$.

Eq. (40): Since it is an approximation, use $\approx$ instead of $=$ here.

p. 14, l. 8: "... the averages of the model $\hat{y}$ (model 1) averages ...": Is this correct?

p. 14, l. 12: Figures should always be given (numbered) in the same order as they appear in the text. Please check.

p. 14, l. 25: "... to avoid rare and unrealistic extreme response of ..." -> "... to avoid rare and unrealistic extreme responses of ..."

p. 14, l. 25: "... (set to 3. adimensional unit) ...": Please check.

p. 21, l. 2: "... which can in some circumstances limit of the applicability ..." -> "... which can in some circumstances limit the applicability ..."

p. 21, l. 6ff: Please revise paragraph.

---

## Author Comment (AC2) · 28 Feb 2020

**1  General comments**

We thank the referee for his/her nice review. We have tried to address his/her recommendations to improve the readability of the article.

**2 Specific comments**

- **The information in Eq. (13)-(14) and (16)-(17) is very redundant, since one only has to exchange the parameters. I suggest to compress this a little bit.**

We understand the comment of the referee to avoid the redundancy of equations (13)-(14) and (16)-(17). However we do believe that this redundancy is necessary at this stage of the article, such that the reader understands clearly the method proposed here (impact of model change on post-processing). It is the critical point where the design of the experiments is clarified, and we do not want to loose the reader at this stage with too compact notations.

- **The application of the concept to the OU-processes is quite helpful. I would suggest to first formulate the response theory in general, and then in a separate section apply and explain the formulas for the OU case, followed by the post-processing.**

  **and also**

- **The application of the response theory here is very not very clear to me, and I still do not quite see, where and how the tangent model comes into play.**

We concur with the reviewer concerning these points, and indeed it is a good suggestion to present the response theory first. We thus added a new section to introduce this technique. We hope that it will be clearer now to a broader audience.

[Figure]

- **It would further be helpful to maybe see a figure of $\alpha(\tau)$ and $\beta(\tau)$.**

  We added a figure showing $\alpha$ and $\beta$.

- **With respect to the QG model it would be helpful to, on the one hand, see the model equations probably in the Fourier space, but, on the other hand, simplify the Fourier expansion (e.g. using complex expansion), since this is a quite standard Galerkin approach (you have to mention the boundary conditions).**

  With respect to the QG model, we prefer to follow the non-complex expansion, since it is the one used in the papers we cite, and it is also how the model was coded. We have provided information on the boundary conditions:

  > "Both fields are defined in a zonally periodic channel with no-flux boundary conditions in the meridional direction ($\partial \cdot /\partial x \equiv 0$ at $y = 0, \pi$). The fields are expanded in Fourier modes respecting these boundary conditions:..."

- **Regarding Fig. 2 (b) you may guide the reader a little bit, e.g., mention, that $\psi_2$ and $\psi_3$ represent the strength of the zonal wavenumber 1 anomalies, which are both small in the dashed line ellipse, and large in the dashed-dot ellipse, which also seems to have a strong zonal mean $\psi_1$ component.**

  We have added more explanations on Fig.2 according to the suggestion of the referee.

- **What is the relevance of the equilibrium ploints?**

  The equilibrium points have no other utility than to help the reader have an idea about the general structure of the global attractor.

**3  Technical corrections**

In general, we agree with most of the corrections proposed by the referee. However, for some corrections, we have additional comments:

**p. 1, l.5 "averages involved": I only understood after reading the paper, what is meant by this.**

We have replaced "averages" by "parameters". This is indeed the parameters that contain the averages of the observables.

**p. 1, l. 9 What is "in a more operational environment"? Maybe "in an environment more similar to the operational environment"**

We changed the sentence into:

" The potential application in a operational environment is also discussed."

**p 3., l. 17 square bracket "⟨⟩" are not explained here, only later on. In general I find it confusing to refer here to averages, as these are to my understanding expectation values (which are indeed estimated via ensemble averages).**

We understand your comment and the confusion that may arise, but we prefer to keep this notation as it is a traditional one in the context of stochastic process and response theory.

**p. 3, l. 29 "Note that if $\kappa = 1$, the correction is perfect.": Not absolutely true, since forecast and model still differ in $\lambda$.**

Thank you very much, the text has been changed to

"Note that the best correction is obtained if $\kappa = 1$."

**p. 6, l. 19 "It is also assumed that there is no interference due to initial condition errors in the problem.": Please clarify, what you mean by this.**

We have removed the sentence, but an explanation is now provided in the new section 2 regarding response theory:

"It is thus assumed that there is no interference due to initial condition errors in the perturbation problem, but note that the effects on the trajectories of both the initial conditions perturbation and the $\boldsymbol{\Psi}$ perturbation can be studied through this equation, by setting $\boldsymbol{\delta y}(0) \neq 0$..."

**Eq. (23) $f^\tau()$ is only given later and specific for the OU process. Please state here, what this is.**

The definition is given just after in the text and Eq. (24). We felt that it was less artificial to introduce it right after its first appearance than before. However, the concept of dynamical system's flow is now also introduced in the new (and anterior) section 2.

**Also Eq.(23) The term in the square brackes may be a scalar product in general, but as you have formulated $\psi_y$, it is not. Later on you use bold letters for vectors, right?**

It is a simple multiplication, and thus we have removed the $\cdot$. Vectors are bold letters, as recommended by the NPG guidelines.

**p. 7, l. 1 "its stochastic integrals ..." > This may need some explanation.**

We think that the paper is already long enough so we prefer to avoid an
introduction about stochastic integrals, but we now cite there Gardiner's book
which is a reference on the subject.

**Eq. (28) Check formula: Isn't there missing $A(f^\tau(x_o))$ and $A(f^\tau(x_o))$ should be
$A(f^\tau(x_o))$? Again is this really a scalar product?**
Formula is right. This is a second-order formula in the perturbation $\Psi$ which
appears twice. On the other hand, the observable $A$ should appears only once
and the final time $\tau$ is correct. We comment on this now in a footnote. Please
see Lucarini (2012) for more details. As before, you are right about the fact that
it is not a scalar product and we removed the $\cdot$.

**p.8, l.8+9 I assume, that the $n$ in the sine and cosine should not be there.**

$n$ is the aspect ratio of the model domain which is $(0 \leq x \leq \frac{2\pi}{n}, 0 \leq y \leq \pi)$, it has
thus to be included in the cosine and sine of $x$. We clarify now the domain in the
model introduction:

"The horizontal adimensionalised coordinates are denoted $x$ and $y$, the model's
domain being defined by $(0 \leq x \leq \frac{2\pi}{n}, 0 \leq y \leq \pi)$, with $n = 2L_y/L_x$ the aspect
ratio between its meridional and zonal extents $L_y$ and $L_x$."

**p. 12, Eq. (39) Use either scalar product or transpose $^T$ .**

The transpose indicates that a row vector is used (rather than a column
one), and the $\cdot$ indicates the kind of product that is being taken (here the
scalar/matricial product indeed). This notation is quite standard in physics (see
Gaspard (2005)), this is ultimately a matter of convention and we want to keep it
the way it is.

**Eq. (40) Since it is an approximation, use $\approx$ instead of $=$ here.**

The referee is right that this is an approximation. But we make the quite standard abuse of notation here that $\delta x$ is directly replaced by its first order approximation. We mention that it is the first order equation. With this abuse of notation, Eq. (40) is an equality and as it is an ODE, we prefer it like that.

**p.21, l. 6ff Please revise paragraph.**

We have changed the paragraph into:

"To test this approach, we have focused on the EVMOS statistical post-processing method, but other methods could be considered as well. The only requirement is that the outcome of the minimisation of the cost function uses averages of the systems being considered. For instance, member-by-member methods that correct both the mean square errors and the spread of the ensemble while preserving the spatial correlation (...) could be considered. These methods generally use the covariance between the model forecasts and the observations as ingredient. Response theory can also be applied here since this covariance can be written as an average. This will be investigated in a future work, together with the applicability of the approach to parameters of probability distributions, as often used in meteorology (...)."

---

## Author Response (AR1)

Dear Editor, Dear Referees,

 In the process of correcting the manuscript, we realized that the model qgs with which the numerical results were obtained contained small errors. These errors do not impact the dynamical core but are related to the orography and temperature profile. As such, by changing the model's parameter we have recovered a model's regime similar to what we previously had. We have performed the analysis again and found no qualitative differences with the previous one, underlying the robustness of our method. In fact, the method shows a good correction of the postprocessing up to 4 days instead of 3. This is due to the atmosphere being less unstable, and the Lyapunov time being slightly longer.

We have updated the figures and the manuscript with these new results. Note that we have also removed the figure 11 of the initial manuscript, since we concluded it has little relevance to the subject of the present manuscript.

We are now confident that the model is correct and bug-free.

We join our updated responses to the referees to this letter. The full list of changes to the manuscript as a latex diff pdf is also joined to this letter.

Sincerely yours,

Jonathan Demaeyer and Stéphane Vannitsem.

**1 Response to referee 1**

We thank the referee for his nice and constructive review and address its comments below:

**1.1 General comments**

We are glad that the referee recognize the potential of this approach. We note that indeed it would be better to already have a realistic application. This article was limited to the proof-of-concept part. The application to real cases is currently considered, but it might require some time before it is done. As such, we hope that the present publication will provide some guidelines and attract interest to work on this. We address now the specific comments:

1. **since this approach supposes the changes in the model are small, we can wonder if it will really be useful for operations, where small changes in the model may not really impair post-processing. Furthermore, it should be interesting to see whether this approach improves over the use of dynamic filtering of residual biases (via Kalman filtering, classically). Do you have any hint about this alternative?**

   In the present model considered, we note that small changes can really impair the postprocessing. See now Fig. 9(a) and 10(a). Furthermore, experiments showed that in some cases, linear response theory can be valid outside of its predicted range of validity (see Gottwald et al. (2016) and Wormell and Gottwald (2018)). Dynamic filtering could be corrected as well, it is a good point. For instance by adjusting the past predictions in the filtering window to the new model via the response theory presented here.

2. **the stated conditions for using this correcting seem very strong: a tangent model must be available, the model change has to be provided as an analytic function. These two conditions may not be observed or the models themselves may not be available. Would it be possible to follow the same procedure in a data-based approach? In other words, could it be possible to deduce the necessary correction if one has only the two sets of forecasts on a common period without access to the models themselves?**

   A tangent model is often available together with the forecasting model, at least to evaluate the sensivity to initial condition. It would cost less to develop than a reforecasting system, and several out-of-the-box automatic differentiation packages also exist to do it. At this stage of the research, we do not see how to apply the method in a purely data-driven context. This is indeed worth exploring in the future.

3. **the extend of post-processing methods that may be corrected in this way is not very clear. You state that The only requirement is that the outcome of the minimization of the cost functions uses averages of the systems being considered. Does this mean that post-processing methods that do not use cost functions (such as random forest) are not eligible to this approach?**

We do not have expertise on methods such as random forest, but in any case it is an interesting question that could be addressed in a subsequent work. For instance, the present method could maybe be used in conjunction with random forest, using directly the trajectories of the tangent model instead of the reforecasts trajectories. The caveat here is then to deal with the problem of fat tails. Indeed, some trajectories (the ones forming the fat tails) will show extreme deviations from the others. So new methods should be designed to take care of this issue.

**1.2 Specific comments**

1. **Do you have a reference to support this claim that post-processing is useful only up to a lead time of 3 days?**

We believe that figures 9(a) and 10(a) are sufficient to support that claim. The claim concerns only the highly truncated system at hand, and not the current state-of-the-art models.

2. **Could you develop about what kind of sophistication you are thinking about?**

We are mainly thinking about other MOS schemes, but as said above, other kind of postprocessing could be considered (Random Forest, Machine Learning, ...). We are more specific in the text now, and write (page 7, line 14 of the final manuscript):

> "More sophisticated approaches can be evaluated in the future (other MOS schemes, ensemble MOS, ...)."

3. **For the article to be self-contained, may you add a more comprehensive definition (with equations) for this system, maybe in the appendix?**

The equations of the model have been added in an appendix (Appendix B, pages 28-29 of the final manuscript).

4. **Is there a simple explanation why your system has only one blocking regime instead of two?**

To our knowledge, it is not easy to explain intuitively such regime shift in nonlinear systems. A full understanding of the phenomenon should

include a bifurcation analysis, and the analysis should then be translated to a physical description. In short, we do not have such a simple explanation for now.

5. **These pictures are not very clear. In panel (b) only model 0 seems to have two attractors but from the text (page9, line 12 and elsewhere) I understand that reality also should have two attractors. Please can you clarify either the text or the pictures?**

   The referee is right, the global attractor of both reality and model 0 contains two parts. It is now mentioned in the figure 2 caption :

   "The attractors of the reality and model 0 are qualitatively similar, with two different parts which are indicated by ellipses."

6. **Later on, you say that the distribution of perturbation has been approximated with a gaussian distribution. In the conclusion, you propose to use the CLV method to get better corrections at farther lead times. I was wondering if we could improve the correction at long lead times by using a different distribution (with fatter tails) to approximate the perturbation distribution?**

   This is a possibility, but to our knowledge, estimating moments of fat tails distributions is quite difficult. This can be considered in a future work, together with the development of the approach in a more realistic environment.

7. **Based on the results, it seems this approach may require a very short period where both models are available. The length of the comon period (a few months?) seems to correspond to what may be available operationnaly in national weather services. This is a very good point, to check on real data, maybe in a future study.**

   There is no need of an overlapping period, except for verification of the method. In the future we indeed plan to use this approach in a realistic context.

**1.3 Technical corrections**

All the technical corrections have been addressed. We thank again the reviewer for his help.

**2 Response to referees 2**

**2.1 General comments**

We thank the referee for his/her nice review. We have tried to address his/her recommendations to improve the readability of the article.

**2.2 Specific comments**

- **The information in Eq. (13)-(14) and (16)-(17) is very redundant, since one only has to exchange the parameters. I suggest to compress this a little bit.**

  We understand the comment of the referee to avoid the redundancy of equations (13)-(14) and (16)-(17). However we do believe that this redundancy is necessary at this stage of the article, such that the reader understands clearly the method proposed here (impact of model change on post-processing). It is the critical point where the design of the experiments is clarified, and we do not want to loose the reader at this stage with too compact notations.

- **The application of the concept to the OU-processes is quite helpful. I would suggest to first formulate the response theory in general, and then in a separate section apply and explain the formulas for the OU case, followed by the post-processing.**

  **and also**

- **The application of the response theory here is very not very clear to me, and I still do not quite see, where and how the tangent model comes into play.**

  We concur with the reviewer concerning these points, and indeed it is a good suggestion to present the response theory first. We thus added a new section (Section 2, pages 3 to 5 of the final manuscript) to introduce this technique. We hope that it will be clearer now to a broader audience.

- **It would further be helpful to maybe see a figure of $\alpha(\tau)$ and $\beta(\tau)$.**

  We added a figure showing $\alpha$ and $\beta$ (figure 8, page 21 of the final manuscript).

- **With respect to the QG model it would be helpful to, on the one hand, see the model equations probably in the Fourier space, but, on the other hand, simplify the Fourier expansion (e.g. using complex expansion), since this is a quite standard Galerkin approach (you have to mention the boundary conditions).**

With respect to the QG model, we prefer to follow the non-complex expansion, since it is the one used in the papers we cite, and it is also how the model was coded. We have provided information on the boundary conditions (page 11, lines 4-5 of the final manuscript):

"Both fields are defined in a zonally periodic channel with no-flux boundary conditions in the meridional direction ($\partial \cdot /\partial x \equiv 0$ at $y = 0, \pi$). The fields are expanded in Fourier modes respecting these boundary conditions:..."

- **Regarding Fig. 2 (b) you may guide the reader a little bit, e.g., mention, that $\psi_2$ and $\psi_3$ represent the strength of the zonal wavenumber 1 anomalies, which are both small in the dashed line ellipse, and large in the dashed-dot ellipse, which also seems to have a strong zonal mean $\psi_1$ component.**

We have added more explanations on Fig.2 according to the suggestion of the referee (page 12 last line, page 13 first line of the final manuscript):

"In the former case, the variables $\psi_2$ and $\psi_3$ characterising the strength of the meridional anomalies are small, while in the latter case they are large, indicating indeed a blocking situation."

- **What is the relevance of the equilibrium ploints?**

The equilibrium points have no other utility than to help the reader have an idea about the general structure of the global attractor.

**2.3  Technical corrections**

In general, we agree with most of the corrections proposed by the referee. However, for some corrections, we have additional comments:

**p. 1, l.5 "averages involved": I only understood after reading the paper, what is meant by this.**

We have replaced "averages" by "parameters". This is indeed the parameters that contain the averages of the observables.

**p. 1, l. 9 What is "in a more operational environment"? Maybe "in an environment more similar to the operational environment"**

We changed the sentence into (last line of the abstract of the final manuscript):

" The potential application in a operational environment is also discussed."

**p 3., l. 17 square bracket "$\langle \rangle$" are not explained here, only later on. In general I find it confusing to refer here to averages, as these are to my understanding expectation values (which are indeed estimated via ensemble averages).**

We understand your comment and the confusion that may arise, but we prefer to keep this notation as it is a traditional one in the context of stochastic process and response theory.

**p. 3, l. 29 "Note that if $\kappa = 1$, the correction is perfect.": Not absolutely true, since forecast and model still differ in $\lambda$.**

Thank you very much, the text has been changed to (page 6 line 20 of the new manuscript):

"Note that the best correction is obtained if $\kappa = 1$."

**p. 6, l. 19 "It is also assumed that there is no interference due to initial condition errors in the problem.": Please clarify, what you mean by this.**

We have removed the sentence, but an explanation is now provided in the new section 2 regarding response theory (last line of page 4 of the final manuscript):

"It is thus assumed that there is no interference due to initial condition errors in the perturbation problem. Note however that the effects on the trajectories of both the initial conditions perturbation and the $\boldsymbol{\Psi}$ perturbation can be investigated through this equation by setting $\boldsymbol{\delta y}(0) \neq 0$, although we are not aware of any study of the response to both type of perturbations together."

**Eq. (23) $f^\tau()$ is only given later and specific for the OU process. Please state here, what this is.**

The definition is given just after in the text and Eq. (24). We felt that it was less artificial to introduce it right after its first appearance than before. However, the concept of dynamical system's flow is now also introduced in the new (and anterior) section 2.

**Also Eq.(23) The term in the square brackes may be a scalar product in general, but as you have formulated $\psi_y$, it is not. Later on you use bold letters for vectors, right?**

It is a simple multiplication, and thus we have removed the $\cdot$. Vectors are bold letters, as recommended by the NPG guidelines.

**p. 7, l. 1 "its stochastic integrals ..." > This may need some explanation.**

We think that the paper is already long enough so we prefer to avoid an introduction about stochastic integrals, but we now cite there Gardiner's book which is a reference on the subject (page 9, line 22 of the final manuscript).

**Eq. (28) Check formula: Isnt there missing $A(f^\tau(x_o))$ and $A(f^\tau(x_o))$ should be $A(f^\tau(x_o))$? Again is this really a scalar product?**
Formula is right. This is a second-order formula in the perturbation $\boldsymbol{\Psi}$ which appears twice. On the other hand, the observable $A$ should appears only once and the final time $\tau$ is correct. We comment on this now in a footnote. Please see Lucarini (2012) for more details. As before, you are right about the fact that it is not a scalar product and we removed the $\cdot$.

**p.8, l.8+9 I assume, that the $n$ in the sine and cosine should not be there.**

$n$ is the aspect ratio of the model domain which is
$(0 \leq x \leq \frac{2\pi}{n}, 0 \leq y \leq \pi)$, therefore it must be included in the cosine and sine of $x$. We clarify now the domain in the model introduction (2 first lines of page 11 of the final manuscript):

"The horizontal adimensionalised coordinates are denoted $x$ and $y$, the model's domain being defined by $(0 \leq x \leq \frac{2\pi}{n}, 0 \leq y \leq \pi)$, with $n = 2L_y/L_x$ the aspect ratio between its meridional and zonal extents $L_y$ and $L_x$."

**p. 12, Eq. (39) Use either scalar product or transpose $^T$.**

The transpose indicates that a row vector is used (rather than a column one), and the $\cdot$ indicates the kind of product that is being taken (here the scalar/matricial product indeed). This notation is quite standard in physics (see Gaspard (2005)), this is ultimately a matter of convention and we want to keep it the way it is.

**Eq. (40) Since it is an approximation, use $\approx$ instead of $=$ here.**

The referee is right that this is an approximation. But we make the quite standard abuse of notation here that $\boldsymbol{\delta x}$ is directly replaced by its first order approximation. We mention that it is the first order equation. With this abuse of notation, Eq. (40) is an equality and as it is an ODE, we prefer it like that.

**p.21, l. 6ff Please revise paragraph.**

We have changed the paragraph into:

[revised manuscript text omitted]

---

## Referee Report (RR1)

**General comments**

The authors have addressed my previous comments. I have still a few (minor) comments (typos, wrong references, some clarification, ...).

  I thus recommend this article for minor revision, so that it be published after the comments have been taken into account. As said in the previous review, it will be a valuable contribution to the litterature about statistical postprocessing of weather forecasts.

**Specific comments**

Hereafter, the passages quoted from the article are in italics, whereas my comments are in normal font.

1. *page 1, line 9: The potential application in a operational environment*
   Please change into "The potential application in an operational environment".

2. *page 3, line 28: Without loss of generality, we shall assume for simplicity that the system (1) is autonomous*
   How this hypothesis can be 'without loss of generality'? It seems rather a strong hypothesis to a non specialist of dynamic system like me. Please explain.

3. *page 12, line 6: the only non-zero coefficients are $\theta_1^\star = 0, 2$*
   Table 1 gives a value for $\theta_2^\star$. I guess the table is wrong, please correct.

4. *page 12, line 11: In particular the system possesses two distinct weather regimes, depicted in Fig. 2(b):*
   Fig 2 (a) and (b) are hard to read and hardly informative (no obvious structure appears in the scatterplots). I would suggest improving the readability or removing the subfigures. For the 2D scatterplot, maybe plotting isodensity lines for the model and the reality on the same subfigure would be more informative.

5. *page 12, line 13: In particular the system possesses two distinct weather regimes, depicted in Fig. 2(b): one characterised by a zonal circulation (see Fig. 2(c)), and another characterised by a blocking situation (see Fig. 2(d)).*
   I would say that the regimes are reversed in your figures: zonal circulation in Fig 2(d) and blocking situation in Fig 2(c).

6. *page 12, line 1: time evolution of the variable $\psi_4$*
   In Fig. 1 (b), the depicted variable is $\psi_2$, not $\psi_4$. Please correct.

7. *page 17, line 24: As it can be seen in Fig. 7,*
   Please change into "As it can be seen in Fig. 7 for the perturbations on $\theta_1$,".

8. *page 17, line 30: The moments obtained by the response theory approach are used to compute new EVMOS postprocessing $\alpha$ and $\beta$*

   *coefficients, thanks to the formulas (17) and (18).*
   Shouldn't you refer to Eqs 28 and 29?

9. *page 17, line 31: These corrected coefficients are shown in Fig. 8 and in the panels (c) and (d) of Fig. 11.*
   Please change into "These corrected coefficients for variable $\theta_1$ are shown in Fig. 8 for the experiment varying the Newtonian cooling coefficient and in the panels (c) and (d) of Fig. 11 for the experiment varying the friction coefficient.".

10. *page 18, line 7: In the panels (b) and (c) of Figs. 9 and 10, the mean and variance of the corrected forecasts is compared with those of the original models. Again, these corrections are efficient until 4 days for the postprocessing schemes*
    I don't see this: to me, the mean and variance of the post-processed forecasts for variable $\theta_1$ seem almost perfect at all lead times. This is an apparent discrepancy with the conclusions drawn from Fig. 9: page 18, line 5, you rightfully notice from the evolution of the MSE with lead time that *the statistical postprocessing corrections are efficient until lead times of 4-5 days.*. It should be explained how the MSE for variable $\theta_1$ can be improved only up to 4 days ahead while the first two moments are perfectly corrected at all lead times. Higher moments of variable $\theta_1$ may explain, at least partially, this discrepancy, along with the temporality of the forecasts.

11. *page 18, line 12: the variance needed to compute the $\alpha$ and $\beta$*

    *coefficients (see Eqs. (17) and (18)).*
    Shouldn't you refer to Eqs 28 and 29?

12. *page 20, line 1: Figure 7. Histograms of the solutions of the equation (50) for the perturbation $\delta y(\tau)$*
    Please change into "Figure 7. Histograms of the solutions of the equation (50) for the perturbation $\delta y(\tau)$ (with $y = \theta_1$)".

13. *page 21, line 1: Figure 8. Coefficients $\alpha$ and $\beta$*

    *of the postprocessing schemes*
    Please change into "Figure 8. Coefficients $\alpha$ and $\beta$ of the postprocessing schemes of variable $\theta_1$".

14. *page 24, line 1: Fig 11*
    Please add missing captions a), b), c) and d) to the subfigures.

    Michaël Zamo, Météo-France.

---

## Editor Decision (ED1)

Review of Manuscript npg-2019-57

Correcting for Model Changes in Statistical Postprocessing – An approach based on Response Theory

**General comments**

The authors have addressed my previous comments. I have still a few (minor) comments (typos, wrong references, some clarification, ...).

I thus recommend this article for minor revision, so that it be published after the comments have been taken into account. As said in the previous review, it will be a valuable contribution to the litterature about statistical postprocessing of weather forecasts.

**Specific comments**

Hereafter, the passages quoted from the article are in italics, whereas my comments are in normal font.

- 1. page 1, line 9: The potential application in a operational environment Please change into "The potential application in an operational environment".
- 2. page 3, line 28: Without loss of generality, we shall assume for simplicity that the system (1) is autonomous
  How this hypothesis can be 'without loss of generality'? It seems rather a strong hypothesis to a non specialist of dynamic system like me. Please explain.
- 3. page 12, line 6: the only non-zero coefficients are  $\theta_1^{\star} = 0, 2$ Table 1 gives a value for  $\theta_2^{\star}$ . I guess the table is wrong, please correct.
- 4. page 12, line 11: In particular the system possesses two distinct weather regimes, depicted in Fig. 2(b):
  Fig 2 (a) and (b) are hard to read and hardly informative (no obvious structure appears in the scatterplots). I would suggest improving the readability or removing the subfigures. For the 2D scatterplot, maybe plotting isodensity lines for the model and the reality on the same subfigure would be more informative.
- page 12, line 13: In particular the system possesses two distinct weather regimes, depicted in Fig. 2(b): one characterised by a zonal circulation (see Fig. 2(c)), and another characterised by a blocking situation (see Fig. 2(d)).

I would say that the regimes are reversed in your figures: zonal circulation in Fig 2(d) and blocking situation in Fig 2(c).

- 6. page 12, line 1: time evolution of the variable  $\psi_4$ In Fig. 1 (b), the depicted variable is  $\psi_2$ , not  $\psi_4$ . Please correct.
- 7. page 17, line 24: As it can be seen in Fig. 7, Please change into "As it can be seen in Fig. 7 for the perturbations on  $\theta_1$ ,".

Correcting for Model Changes in Statistical Postprocessing – An approach based on Response Theory

 page 17, line 30: The moments obtained by the response theory approach are used to compute new EVMOS postprocessing α and β

coefficients, thanks to the formulas (17) and (18). Shouldn't you refer to Eqs 28 and 29?

- 9. page 17, line 31: These corrected coefficients are shown in Fig. 8 and in the panels (c) and (d) of Fig. 11.
  Please change into "These corrected coefficients for variable θ1 are shown in Fig. 8 for the experiment varying the Newtonian cooling coefficient and in the panels (c) and (d) of Fig. 11 for the experiment varying the friction coefficient.".
- 10. page 18, line 7: In the panels (b) and (c) of Figs. 9 and 10, the mean and variance of the corrected forecasts is compared with those of the original models. Again, these corrections are efficient until 4 days for the postprocessing schemes

I don't see this: to me, the mean and variance of the post-processed forecasts for variable  $\theta_1$  seem almost perfect at all lead times. This is an apparent discrepancy with the conclusions drawn from Fig. 9: page 18, line 5, you rightfully notice from the evolution of the MSE with lead time that the statistical postprocessing corrections are efficient until lead times of 4-5 days.. It should be explained how the MSE for variable  $\theta_1$  can be improved only up to 4 days ahead while the first two moments are perfectly corrected at all lead times. Higher moments of variable  $\theta_1$  may explain, at least partially, this discrepancy, along with the temporality of the forecasts.

- page 18, line 12: the variance needed to compute the α and β coefficients (see Eqs. (17) and (18)). Shouldn't you refer to Eqs 28 and 29?
- 12. page 20, line 1: Figure 7. Histograms of the solutions of the equation (50) for the perturbation δy(τ)
  Please change into "Figure 7. Histograms of the solutions of the equation (50) for the perturbation δy(τ) (with y = θ1)".
- 13. page 21, line 1: Figure 8. Coefficients  $\alpha$  and  $\beta$

of the postprocessing schemes Please change into "Figure 8. Coefficients  $\alpha$  and  $\beta$  of the postprocessing schemes of variable  $\theta_1$ ".

14. page 24, line 1: Fig 11 Please add missing captions a), b), c) and d) to the subfigures.

Michaël Zamo, Météo-France.

---

## Author Response (AR2)

Dear Editor, Dear Referees,

We have modified the text and figures according to the modifications proposed by the referee. In particular, figure 2 has been reworked to depict the attractor structure more clearly.

We join our response to the referee comments to this letter. The full list of changes to the manuscript as a latex diff pdf is also joined to this letter.

Sincerely yours,

Jonathan Demaeyer and Stéphane Vannitsem.

**1 Response to the referee comments**

We thank warmfully the referee for his detailed review of the manuscript. We address its comments below:

**1.1 Specific comments**

1. **page 1, line 9:** *The potential application in a operational environment*
   **Please change into The potential application in an operational environment.**

   Done. Thank you.

2. **page 3, line 28:** *Without loss of generality, we shall assume for simplicity that the system (1) is autonomous*
   **How this hypothesis can be without loss of generality ? It seems rather a strong hypothesis to a non specialist of dynamic system like me. Please explain.**

   The formal description is in principle valid for non-autonomous systems. In the present context we are dealing with an autonomous one, so the theory is developed for such type of system. We removed the begining of the sentence and now state:

   > "We shall assume that the system (1) is autonomous"

   at the line 28, page 3 of the new manuscript.

3. **page 12, line 6:** *the only non-zero coefficients are $\theta_1^\star = 0.2$.*
   **Table 1 gives a value for $\theta_2^\star$. I guess the table is wrong, please correct.**

   The referee is right, Table 1 has been changed. Thank you.

4. **page 12, line 11:** *In particular the system possesses two distinct weather regimes, depicted in Fig. 2(b):*
   **Fig 2 (a) and (b) are hard to read and hardly informative (no obvious structure appears in the scatterplots). I would suggest improving the readability or removing the subfigures. For the 2D scatterplot, maybe plotting isodensity lines for the model and the reality on the same subfigure would be more informative.**

   We have replaced the 3D scatter plot by a 2D isodensity plot matching the Fig. 2(b) 2D scatter plot. We have also added the fixed points of both reality and model 0. The presentation is now much better. Thank you.

5. **page 12, line 13:** *In particular the system possesses two distinct weather regimes, depicted in Fig. 2(b): one characterised by a zonal circulation (see Fig. 2(c)), and another characterised by a blocking situation (see Fig. 2(d)).* **I would say that the regimes are reversed in your figures: zonal circulation in Fig 2(d) and blocking situation in Fig 2(c).**

   In the process of refactoring the notebooks code, the two figures Figs. 2(c) and 2(d) had indeed been permuted. We have now repermuted them back. Thank you.

6. **page 12, line 1:** *time evolution of the variable $\psi_4$* **In Fig. 1 (b), the depicted variable is $\psi_2$, not $\psi_4$ . Please correct.**

   Ok. Thank you.

7. **page 17, line 24:** *As it can be seen in Fig. 7,* **Please change into As it can be seen in Fig. 7 for the perturbations on $\theta_1$ ,.**

   Thank you very much.

8. **page 17, line 30:** *The moments obtained by the response theory approach are used to compute new EVMOS postprocessing $\alpha$ and $\beta$ coefficients, thanks to the formulas (17) and (18).* **Shouldnt you refer to Eqs 28 and 29?**

   No, we use directly the formulas (17) and (18) with the moments correction obtained directly after applying response theory.

9. **page 17, line 31:** *These corrected coefficients are shown in Fig. 8 and in the panels (c) and (d) of Fig. 11.* **Please change into These corrected coefficients for variable 1 are shown in Fig. 8 for the experiment varying the Newtonian cooling coefficient and in the panels (c) and (d) of Fig. 11 for the experiment varying the friction coefficient..**

   Thank you very much for the suggestion. We have changed the text as follow (page 17, lines 31 to 33 of the new manuscript):

   "These corrected coefficients for variable $\theta_1$ are shown in Fig. 8 for the experiment with modification of the Newtonian cooling coefficient and in the panels (c) and (d) of Fig. 11 for the experiment with modification of the friction coefficient."

10. **page 18, line 7:** *In the panels (b) and (c) of Figs. 9 and 10, the mean and variance of the corrected forecasts is compared*

*with those of the original models. Again, these corrections are efficient until 4 days for the postpro- cessing schemes*
**I dont see this: to me, the mean and variance of the post-processed fore- casts for variable 1 seem almost perfect at all lead times. This is an apparent discrepancy with the conclusions drawn from Fig. 9: page 18, line 5, you rightfully notice from the evolution of the MSE with lead time that the statistical postprocessing corrections are efficient until lead times of 4-5 days.. It should be explained how the MSE for variable 1 can be improved only up to 4 days ahead while the first two moments are perfectly corrected at all lead times. Higher moments of variable 1 may explain, at least partially, this discrepancy, along with the temporality of the forecasts.**

The correction of the 2 first moments is independent of the mechanism leading to the increase of the MSE over time, which is due to the dynamics of the model error. EVMOS is able to correct part of the error up to 2-3 times the Lyapunov time (the inverse of the leading Lyapunov exponent) of the model. After that the two MSE curves (for raw and corrected forecasts) converge toward a plateau which is simply twice the variance of the reference solutions. This is the way the EVMOS is designed. We have added details about that and modified the end of the "Main results" section 4.3 . The text has been modified as follows (see page 18 lines 4 to 14 of the new manuscript):

"The EVMOS postprocessing is able to partly correct the forecasts, reducing the MSE until a lead time of the order of a few times the model's Lyapunov time (the inverse of the leading Lyapunov exponent). After that, the MSE curves of the postprocessed and uncorrected forecasts converge toward a plateau corresponding to twice the variance of the reference solution (...). Here, the statistical postprocessing corrections are indeed efficient until lead times of 4-5 days, with a skill of the corrections decreasing with the lead time. Thus the EVMOS schemes become not better than the original models after roughly 4 days. Note also that even if the model change is small, the postprocessing using the past forecasts of model 0 (green "+" crosses) completely fails to correct model 1 forecasts, highlighting the need for an adaptation of the postprocessing to the model change. In contrast, the adaptation with the response-theory method (black "×" crosses) produces valid corrections until 4 days ahead. In the panels (b) and (c) of Figs. 9 and 10, the mean and variance of the corrected forecasts are compared with those of the original models. Again, the corrections obtained with response theory are efficient until 4 days for the postprocessing schemes."

11. **page 18, line 12:** *the variance needed to compute the and coefficients (see Eqs. (17) and (18)).* **Shouldnt you refer to**

**Eqs 28 and 29?**

No, please see the reply to the comment 8 above.

12. **page 20, line 1:** *Figure 7. Histograms of the solutions of the equation (50) for the perturbation* $\delta y(\tau)$
    **Please change into Figure 7. Histograms of the solutions of the equation (50) for the perturbation** $\delta y(\tau)$ **(with** $y = \theta_1$**).**

    We modified the text as (page 20, line 1 of the new manuscript):

    > "Histograms of the solutions $\delta\theta_1(\tau)$ of the equation (50) for the perturbation $\boldsymbol{\delta y}(\tau)$ (with $\theta_1 = y_{11}$) along the trajectories of model 0,..."

13. **page 21, line 1:** *Figure 8. Coefficients and of the postprocessing schemes*
    **Please change into Figure 8. Coefficients** $\alpha$ **and** $\beta$ **of the postprocessing schemes of variable** $\theta_1$**.**

    Thank you very much for the suggestion.

14. **page 24, line 1:** *Fig 11*
    **Please add missing captions a), b), c) and d) to the subfigures.**

    Ok. Thank you.

**2 Manuscript differences (latexdiff)**

Please see next page.

[revised manuscript text omitted]